# Cascading Bandits: Optimizing Recommendation Frequency in Delayed Feedback Environments

**Dairui Wang**
Tsinghua University
wdr23@mails.tsinghua.edu.cn

**Junyu Cao**[*]
The University of Texas at Austin
junyu.cao@mccombs.utexas.edu

**Yan Zhang**
McGill University
yan.zhang13@mail.mcgill.ca

**Wei Qi**[*]
Tsinghua University
qiw@tsinghua.edu.cn

## Abstract

Delayed feedback is a critical problem in dynamic recommender systems. In practice, the feedback result often depends on the frequency of recommendation. Most existing online learning literature fails to consider optimization of the recommendation frequency, and regards the reward from each successfully recommended message as equal. In this paper, we consider a novel cascading bandits setting, where individual messages from a selected list are sent to a user periodically. Whenever a user does not like a message, she may abandon the system with a probability positively correlated with the recommendation frequency. A learning agent needs to learn both the underlying message attraction probabilities and users' abandonment probabilities through the randomly delayed feedback. We first show a dynamic programming solution to finding the optimal message sequence in deterministic scenarios, in which the reward is allowed to vary with different messages. Then we propose a polynomial time UCB-based offline learning algorithm, and discuss its performance by characterizing its regret bound. For the online setting, we propose a learning algorithm which allows adaptive content for a given user. Numerical experiment on AmEx dataset confirms the effectiveness of our algorithms.

## 1 Background

### 1.1 Introduction

Recommending dynamic content based on adaptive learning of user behavior is a critical task to online platforms. In marketing campaign, as only 2% of web traffic converts on the first visit[1], it is important to re-target potential users with notifications such as ads, emails or app reminders to gain traction and awareness. The content for those notifications should be dynamic and adapted to a user's behavior. Since the feedback may not be instant, a campaign system still needs to provide campaigns to other users while waiting. Meanwhile, the frequency at which the notifications are sent should also be judiciously managed, i.e., sending more messages may increase the odds of a click, but frequent messaging may also alienate and drive away users.

Online learning for recommendation tasks has been extensively studied in the multi-armed bandit framework. In this setting, pulling certain arms in the bandit problem corresponds to selecting marketing messages for a user. In particular, cascading bandit [13, 14, 18, 21, 20, 25] which models

---

[*]Corresponding authors.
[1]https://retargeter.com/what-is-retargeting-and-how-does-it-work/

sending a sequence of messages to individual users is a popular choice for analyzing sequential recommendations. However, existing settings in this field are often stylized and overlook certain characteristics that are prevalent in real-world scenarios. For instance, a common assumption is immediate user feedback, whereas in practice, it may take hours to months to receive responses. Additionally, bandit models typically disregard the variation in payback for each message, which limits the practical implications of the model for recommendations. Furthermore, the concept of "frequency" in message dissemination is absent in current cascading bandit literature, where an entire sequence of messages is sent to a user at once. Consequently, the content for a single user becomes non-adaptive, potentially missing out on additional feedback from other users over an extended time span and foregoing potential benefits.

In this work, we propose and analyze a variant of cascading bandit with five notable differences from the existing literature. Firstly, whenever a user does not like a message, she may abandon the system and forgo receipt of future messages. Thus, the order of messages influences the reward in our setting, which is in contrast to the existing work on cascading bandit such as [13, 14, 18, 21, 20, 25]. Secondly, instead of having the total number of messages in a sequence as a fixed parameter, this quantity is also a decision variable. In particular, we assume the abandonment probability may increase with the frequency at which messages are being disseminated. Thirdly, individual messages for each user are sent out with some time apart, as opposed to all at once. As we gather more feedback from some of the previous messages, we have the ability to update the remaining sequence, and thus the content for a user is dynamic and adaptive. Fourthly, the reward generated after a message is clicked varies with the type of it. Lastly, we also incorporate delayed feedback in our setting.

The analysis for this setting is particularly challenging. One source of complications comes from the interlaced timing of multiple events, i.e., users arrive at different time, messages for individual users are sent at different time, and their responses time is also random. Meanwhile, users' feedback also depends on multiple factors, including the choices of the content, their order, as well as the dissemination frequency.

The contribution of our work is fourfold. 1. We introduce a novel cascading bandit setting with delayed feedback, different reward and frequency control. 2. For the offline combinatorial problem with all known parameters, we characterize a polynomial-time algorithm which determines an optimal list with the appropriate dissemination frequency. 3. For the online setting where we need to learn both the message attraction probabilities and the abandonment probabilities, we propose a learning algorithm and characterized its regret bound. 4. We incorporate personalization by analyzing a contextual variant of the online problem which utilizes both content and user features.

The paper is organized as follows. We review the related literature in Section 1.2. In Section 2, we formally introduce the model and analyze the offline optimization problem with all known parameters. In Section 3 and Section 4, we study the online learning problem in the presence of delayed feedback for the non-contextual and contextual settings, respectively. We evaluate the performance of our algorithms with numerical experiments in Section 5. Lastly, we discuss some limitations and research prospect of our work in Section 6.

## 1.2 Related Work

The analysis on delayed feedback has started to receive increasing attention in recent years [1, 22, 16, 23]. In online learning, [8] considers a fixed and known delay effect, and shows an additive penalty of the regret for the stochastic setting. [11] analyzes a more general setting and show that the regret increase is multiplicative in the adversarial setting and additive in the stochastic setting. They propose meta-algorithms that can adapt non-delay algorithms and develop a UCB algorithm for the bandit problem. [7] and [19] analyze the impact of delayed feedback in a Gaussian process bandit and an online Markov decision process respectively. Compared to the existing bandit literature, to the best of our knowledge, our work is the first to incorporate delay effects for the cascade model which involves a sequence of actions. Besides the uncertainty contributed by the random response time, the feedback also depends on both the choice of the messages and the order in which they are being sent.

Our work shares some similarity with combinatorial bandits [3, 5, 15, 4] as the learning agent needs to choose a set of messages. However, in our setting, the order of the messages also impacts the rewards. As highlighted in the introduction, our work extends the cascading bandits literature by including frequency control and delayed feedback. This setting can be potentially modeled as a reinforcement

learning problem, such as [24], which proposes an offline reinforcement learning framework to optimize sequential notification decisions. However, this approach significantly increases the learning complexity, making it difficult to analyze performance theoretically.

In the original cascading model [6], a user who is presented with a list of messages examines the content sequentially, until she clicks on the first attractive message. A common different assumption is the *dependent click model*. In this model, a user may click on multiple messages and finally stop at the truly attractive one. It can be used in bandit settings [12] or generic click-models algorithms [9]. In cascading bandit [13, 14, 18, 21, 20, 25], the attraction probabilities of messages are not known a priori. A learning agent selects a list with a fixed number of messages to incoming users, with the goal to maximize the total clicks over $T$ time steps. User's feedback to a list is immediate. The agent then uses this information to learn the underlying attraction probabilities and updates their list for the next user. Although [10] provides a bound based on Thompson sampling for semi-bandit rewards and [20] provides a problem-independent bound for unstructured rewards, they do not consider the messages' difference in theory. While [2] also considers user abandonment behavior in a cascading bandit setting, the abandonment probability is a constant and the recommendation to a single user is static, as opposed an adaptive sequence which is considered in this work.

## 2 Problem Formulation

Assume there are $N$ available messages, which we denote as $X$. We use $\mathbf{S} = (S_1, S_2, \cdots, S_m)$ to denote a list of messages, where $S_1$ and $S_m$ represent the first and the last message, respectively. Let $\kappa$ be the index function mapping the position in a list to a message, i.e., $\kappa(i) = j$ if and only if $S_i = \{j\}$. Define $\vartheta$ as the inverse function of $\kappa$, i.e., $\vartheta(i) = j$ if and only if $S_j = \{i\}$.

Users will receive a list of messages periodically. Upon receiving a message, if a user finds it attractive, she clicks on it. Otherwise, she may examine the next message with probability $q(m)$ if the list has not run out, or she may leave with probability $1 - q(m)$. The probability $q(m)$ depends on the total length of the list or the message dissemination frequency, which we will elaborate next. For each message $i$, we use $v_i$ to denote its attraction probability which captures the relevance of its content to a user, and use $R_i$ to denote its return when a user clicks on it. We define $\mathbf{v} = (v_1, \cdots, v_N)$ and $\mathbf{R} = (R_1, \cdots, R_N)$.

A user arrives at time $t$, where $t = 1, \cdots, T$. We assume that for every user, there is a fixed time window $D$, during which selected messages will be disseminated[2]. Messages are sent sequentially at a fixed time interval with a dissemination frequency $f(m) := \lceil m/D \rceil$. Here we keep the recommendation frequency fixed for every user, which allows us to easily capture the trade-off between the total number of messages that can be disseminated and the probability of abandonment. At every time step, a user receives at most one message. Define $M$ as the maximum number of messages that can be sent during $D$. We assume $q(m)$ is a non-increasing function with $m$. The intuition is that when the frequency of re-targeting is higher, a user is more likely to be overwhelmed and is more prone to abandonment. We denote $\mathbf{q} = (q(1), \cdots, q(M))$. The following example illustrates the interactions between users and messages at different time.

**Example 2.1.** *Figure 1 shows 5 users that arrive at time $t = 1, \cdots, 5$. Actions taken by users are color-coded, i.e., a user clicks on a specific message (orange), has not clicked but remains active (green), and has abandoned the system (yellow). $M_i, t = j$ denotes that message $i$ is sent at time step $j$, e.g., User 2 receives message 10 at $t = 2$ and message 9 at $t = 6$. While the dissemination frequency is determined for a given user upon arrival, it may be different across users, e.g., User 2 and 5 receive messages at a fixed interval of 4 and 6 time steps respectively.*

For a list with $m$ messages, besides the first message which is guaranteed to be viewed, the probability that message $i$ is being examined, denoted as $w_i(m)$, depends on the outcomes of the earlier messages, i.e., $w_i(m) = \prod_{j=1}^{\vartheta(i)-1}((1 - v_{\kappa(j)})q(m))$. It is the probability that the user does not click on any messages shown before $i$ and still remains in the system. Let $U(\mathbf{S}, \mathbf{v}, q(m))$ denote the payoff that the learning agent receives from a list of messages $\mathbf{S}$, when the message attraction probabilities are $\mathbf{v}$

---

[2]$D$ is known as the re-targeting window in the marketing literature. See `https://www.adroll.com/learn-more/retargeting`.

| | t=1 | 2 | 3 | 4 | 5 | 6 | 7 | 8 | 9 | 10 | 11 | 12 | 13 | 14 | 15 | 16 | 17 | 18 | 19 | 20 | 21 | 22 | 23 | ... |
|---|---|---|---|---|---|---|---|---|---|---|---|---|---|---|---|---|---|---|---|---|---|---|---|---|
| User 1 | M1 | | | | M20 | | | | M5 | | | | | | | | | | | | | | | |
| User 2 | | M10 | | | | | M9 | | | | M18 | | | M16 | | | | | | | | | | |
| User 3 | | | | M15 | | | | | M21 | | | | | | | | | | | | | | | |
| User 4 | | | | | M12 | | | | | | | | | | | | | | | | | | | |
| User 5 | | | | | | M17 | | | | | M5 | | | | | | M21 | | | | | | M11 | |

Figure 1: An illustrative example on user's interactions with the learning agent.

and abandonment probability is $1 - q(m)$. The expected payoff is

$$E[U(\mathbf{S}, \mathbf{v}, \mathbf{R}, q(m))] = \sum_{i \in X} w_i(m) R_i v_i.$$

To determine a list of messages with appropriate dissemination frequency, the learning agent needs to solve an optimization problem that maximizes $E[U(\mathbf{S}, \mathbf{v}, \mathbf{R}, q(m))]$, subject to a constraint that $S_i \cap S_j = \emptyset$ for $i \neq j$, indicating no duplicated messages in the list. We use $\mathbf{S}^*$ to denote the optimal solution to the combinatorial optimization problem.

In the following result, we characterize the property of the optimal list $\mathbf{S}^*$. We then propose an algorithm to identify $\mathbf{S}^*$ in the offline setting, i.e., when the message attraction probabilities $\mathbf{v}$ and the probabilities $\mathbf{q}$ describing users' abandonment behavior are known.

**Theorem 2.2.** *In the optimal sequence $\mathbf{S}^*$, the characteristic parameter of messages*

$$\gamma = \frac{vR}{1 - v(1 - q(m))}$$

*are sorted in a descending order.*

**Proof outline:** We prove the statement by contradiction. If $\mathbf{S}$ is not ordered by $\gamma$, we generate another sequence $\mathbf{S}'$ by swapping a pair of adjacent messages in $\mathbf{S}$. We can prove that the expected payoff of $\mathbf{S}'$ is higher than $\mathbf{S}$, which is a contradiction. $\square$

We include complete proof in the supplementary material. Note that while Theorem 2.2 provides a necessary condition for $\mathbf{S}^*$, it does not indicate that the messages with the highest $\gamma$ should be selected. The structure of $E[U(\mathbf{S}, \mathbf{v}, \mathbf{R}, q(m))]$ shows that each message in the list will only influence the contribution of all the messages after it on the total return. Therefore, we use a dynamic programming approach to determine the list backwards.

---

**Algorithm 1:** Determine the optimal sequence $\mathbf{S}^*$ with frequency control.

---

1 Input messages and their $v$, $R$;
2 **for** $m = 1 : M$ **do**
3      Sort messages based on their $\gamma$ in a descending order $S' = (S_1, S_2, \cdots, S_N)$;
4      **for** $l = m - 1 : N - 1$ **do**
5          $W(m-1, l) = \max_{l+1 \leq i \leq N} v_i R_i$; $G(m-1, l) = \mathrm{argmax}_{l+1 \leq i \leq N} v_i R_i$;
6      **end**
7      **for** $k = m - 1 : 1$ **do**
8          **for** $l = k - 1 : N - m + k - 1$ **do**
9              $W(k-1, l) = \max_{l+1 \leq i \leq N-m+k-1}(v_i R_i + (1 - v_i)q(m)W(k, i))$;
10              $G(k-1, l) = \mathrm{argmax}_{l+1 \leq i \leq N-m+k-1}(v_i R_i + (1 - v_i)q(m)W(k, i)) \cup$
                 $G(k, \mathrm{argmax}_{l+1 \leq i \leq N-m+k-1}(v_i R_i + (1 - v_i)q(m)W(k, i)))$
11          **end**
12      **end**
13      Output $W(0, 0)$, $G(0, 0)$.
14 **end**
15 Select the highest $W(0, 0)$ and record the corresponding $m$ and $G(0, 0)$.

---

Algorithm 1 describes the procedure of finding the optimal list $\mathbf{S}^*$ with frequency $f(m^*)$. Messages will be first sorted in a descending order based on $\gamma$. For any feasible $m$, we selected the messages

backwards. The last message must be the one left with the highest expected return $vR$. We define

$$W(k, l) = \sum_{i=k+1}^{m} v_{\kappa(i)} R_{\kappa(i)} \prod_{s=k+1}^{i-1} (1 - v_{\kappa(s)}) q(m), \quad k \leq l \tag{2.1}$$

as the maximum weighted sum of the last $(m - k)$ messages' return, when the $(k + 1)$-th message to be sent is restricted to be after the $l$-th message in $S'$. $G(k, l)$ is to record the messages selected. The transition equation shows the influence of the $(k - 1)$-message on the feasible place for the $k$-th messages in $S'$. Therefore, $G(0, 0)$ is the optimal $m$ messages, and $W(0, 0)$ means total return. The optimal list is the list with the highest expected return $W(0, 0)$ among the $M$ candidates.

## 3  Learning with Delayed Feedback

In the previous section, we have explored the structure of the optimal list when $\mathbf{v}$ and $\mathbf{q}$ are both known. In this section, we turn to the online learning setting with delayed feedback. To differentiate the time stamps, we will use $r$ to indicate the time when a user arrives (i.e., $r = 1, \cdots, T$) and use $t$ to indicate when messages are being sent. The first message for a user $r$ is sent instantaneously at $t = r$, and the subsequent messages are sent at the frequency determined by $f(m)$.

A message sent at time $t$ is *delayed* if its feedback is received after $t$. Let $\tau$ denote the delayed duration. Throughout this analysis with delayed feedback, we assume that $0 \leq \tau \leq \tau_{max}$ with probability 1, i.e., the delay has a finite support independent of the attraction probability. When the feedback is not received after $\tau_{max}$ (e.g., a user stops responding to messages), we assume the user has abandoned. Note that $\tau_{max}$ is not related to the attraction probability, nor does it imply feedback.

Define $\mathbf{S}^r$ as the list of messages for user $r$, where the total message length is $m_r$. We use regret to measure the performance of a learning algorithm, where the regret for a policy $\pi$ is defined as

$$Reg(T) = E_\pi \left[ \sum_{r=1}^{T} U(\mathbf{S}^*, \mathbf{v}, \mathbf{R}, q(m^*)) - U(\mathbf{S}^r, \mathbf{v}, \mathbf{R}, q(m_r)) \right].$$

### 3.1  Learning Algorithm

As the messages are sent sequentially with some time apart, the learning agent may use additional feedback to update the remaining messages. Thus, the content for a user can be dynamic. We will now present our UCB-based learning strategy to achieve this.

We first define our unbiased estimator for $\mathbf{u}$ and $\mathbf{q}$ respectively. Let $T_i(t)$ be the total number of feedback (i.e., sum of clicks and no-clicks) received for message $i$ by time $t$, and let $c_i(t)$ denote the number of clicks for message $i$ by time $t$. Similarly, let $\tilde{T}_m(t)$ denote the total number of no-clicks from users with dissemination frequency $f(m)$, and $b_m(t)$ denote the number of abandoned users with frequency $f(m)$ by time $t$. Let $n_m(t) = \tilde{T}_m(t) - b_m(t)$.

Define the estimator $\hat{v}_{i,t} = c_i(t)/T_i(t)$ and $\hat{q}_t(m) = n_m(t)/\tilde{T}_m(t)$, which are both unbiased. Next, define the upper confidence bound for $\mathbf{v}$ and $\mathbf{q}$ as follows,

$$v_{i,t}^{UCB} = \hat{v}_{i,t} + \sqrt{2 \frac{\log t}{T_i(t)}}, \text{ and } q_t^{UCB}(m) = \hat{q}_t(m) + \sqrt{2 \frac{\log t}{\tilde{T}_m(t)}}. \tag{3.1}$$

The following lemma proves the concentration bound for these estimators.

**Lemma 3.1.** *For any $t$, we have $P\left(v_{i,t}^{UCB} - \sqrt{8 \frac{\log t}{T_i(t)}} < v_i < v_{i,t}^{UCB}\right) \geq 1 - \frac{2}{t^4}$ for all $i \in X$ and* $P\left(q_t^{UCB}(m) - \sqrt{8 \frac{\log t}{\tilde{T}_m(t)}} < q(m) < q_t^{UCB}(m)\right) \geq 1 - \frac{2}{t^4}$ *for all $1 \leq m \leq M$.*

Then we propose Algorithm 2 for our learning task.

---

**Algorithm 2:** An online learning algorithm for cascading bandits with delayed feedback

---

1  **Initialization:** Set $v_{i,0}^{UCB} = 1$ for all $i \in X$, $q_0^{UCB}(m) = 1$ for $m = 1, \cdots, M$; $t = 1$;
2  **while** $t < T$ **do**
3      **for** *Any feedback for some message $i$* **do**
4          |   Update $v_{i,t}^{UCB}$ and $q_t^{UCB}(m)$ according to Equation (3.1);
5      **end**
6      **for** *Any active user $r$ with message scheduled to be sent at time $t$* **do**
7          |   Compute $\mathbf{S}^r = \arg\max_{\mathbf{S}} E[U(\mathbf{S}, \mathbf{v}_{t-1}^{UCB}, \mathbf{R}, q_{t-1}^{UCB}(m_r))]$ according to Algorithm 1;
8          |   Send the first message $e$ in $\mathbf{S}^r \backslash O^r$ to user $r$; $O_r = O_r \cup \{e\}$;
9      **end**
10     Compute $(\mathbf{S}^t, m_t) = \arg\max_{\mathbf{S},m} E[U(\mathbf{S}, \mathbf{v}_{t-1}^{UCB}, \mathbf{R}, q_{t-1}^{UCB}(m))]$ according to Algorithm 1;
11     Offer $S_1^t$ to user $t$ and the dissemination frequency is set to $f(m_t)$; $O_t = S_1^t$;
12 **end**

---

We introduce the following notations. At time $t$, we consider a user $r$ *active* where $r < t$, if she has not clicked on a message or abandoned, and there is a message scheduled to send to her at time $t$. Define $m_r$ as the total number of messages selected for user $r$, where the corresponding abandonment probability is $1 - q(m_r)$ and the frequency is $f(m_r)$. In the recommendation system, $m_r$ is always fixed for a single user and is determined when she enters the system, but it may vary for users entering the system at different times. Define $O_r$ as the set of messages that have already been sent to user $r$.

Algorithm 2 states that at time $t$, the learning agent first updates the UCB estimators for $\mathbf{v}$ and $\mathbf{q}$ based on the available information. For the user that just arrived at time $t$, the agent determines a list $\mathbf{S}^t$ with frequency $f(m_t)$, and sends the first message $S_1^t$. Meanwhile, for an active user $r$ who is scheduled to have the message sent at time $t$, the learning agent determines a new recommending list $\mathbf{S}^r$ with frequency $f(m_r)$ (determined at time $r$), and then sends the first message in the list $\mathbf{S}^r \backslash O^r$.

### 3.2 Regret Analysis

Before presenting the regret bound for Algorithm 2, we want to highlight the challenges behind the analysis. Firstly, we allow dynamic content for active users who remain in the system. Secondly, the feedback of individual messages may be delayed for a random amount of time and the frequency of the message dissemination across users could be different. User feedback depends on both the choice of content and the dissemination frequency, which further complicates the analysis. To address these challenges, we develop a novel proof technique to analyze the regret.

Let $\mathbf{S}^*$ denote the optimal strategy and $\mathbf{S}^r$ denote the sequence for user $r$ under strategy $\pi$. Without loss of generality, assume $\gamma_1 \geq \gamma_2 \geq \cdots \geq \gamma_N$. Define $\mathcal{F}_t$ as the filtration regarding all the information before time $t$. We say $t \in \mathcal{E}_r$ if a message is sent to user $r$ at time $t$.

Lemma 3.2 proves that with the higher attraction probabilities and the lower the abandonment probabilities, the expected payoff is higher. Lemma 3.3 estimates the upper bound regret of a single recommendation under identical reward. Theorem 3.4 gives an upper bound of regret when delayed feedback exists.

**Lemma 3.2.** *Assume $\mathbf{S}^*$ is the optimal list of messages with length $m^*$. Under the condition that $0 \leq \mathbf{v} \leq \mathbf{v}^{UCB}$ and $0 \leq q(m^*) \leq q^{UCB}(m^*)$, we have*

$$E[U(\mathbf{S}^*, \mathbf{v}^{UCB}, \mathbf{R}, q^{UCB}(m^*))] \geq E[U(\mathbf{S}^*, \mathbf{v}, \mathbf{R}, q(m^*))].$$

**Lemma 3.3.** *When all messages have identical reward, for $t \in \mathcal{E}_r$ and any $q' \in (0, 1)$, we have*

$$E_\pi[E[(U(k, \mathbf{v}, q') - U(e_{t,k}^r, \mathbf{v}, q'))1(\mathbf{v}_t^{UCB} \geq \mathbf{v})|\mathcal{F}_{t-1}]]$$

$$\leq E_\pi \left[ E \left[ \left( v_{e_{t,k}^r, t}^{UCB} - v_{e_{t,k}^r} \right) 1(\mathbf{v}_t^{UCB} \geq \mathbf{v})|\mathcal{F}_{t-1} \right] \right],$$

*where $e_{t,k}^r$ is the $k^{th}$ message sent to user $r$ at time $t$.*

**Theorem 3.4** (Performance bound for Algorithm 2)**.** *The expected regret of Algorithm 2 is bounded above by*

$$Reg(T) \leq C_1(N + M^2)\sqrt{T \log T} + C_2 N \tau_{max}$$

*for some constants $C_1$ and $C_2$.*

**Proof Outline**: First we note that

$$
\begin{aligned}
&E_\pi[U(\mathbf{S}^*, \mathbf{v}, \mathbf{R}, q(m^*))] - E_\pi[U(\mathbf{S}^r, \mathbf{v}, \mathbf{R}, q(m_r))] \\
=&E_\pi[U(\mathbf{S}^*, \mathbf{v}, \mathbf{R}, q(m^*)) - U(\mathbf{S}^*, \mathbf{v}, \mathbf{R}, q(m_r))] \\
&+ E_\pi[U(\mathbf{S}^*, \mathbf{v}, \mathbf{R}, q(m_r)) - U(\mathbf{S}^r, \mathbf{v}, \mathbf{R}, q(m_r))]
\end{aligned}
\tag{3.2}
$$

Let $\mathbf{S}_0^r$ denote the initial list for user $r$ when she just arrives, i.e., $\mathbf{S}_0^r$ is the optimal sequence given $\mathbf{v}_{r-1}^{UCB}$ and $\mathbf{q}_{r-1}^{UCB}$. Note that this list may change at a later time when more information becomes available. Define events $B_{i,t} = \{v_{i,t}^{UCB} - \sqrt{8 \log t / T_i(t)} < v_i < v_{i,t}^{UCB}\}$ and $E_{m,t} = \{q_t^{UCB}(m) - \sqrt{8 \log t / \hat{T}_m(t)} < q(m) < q_t^{UCB}(m)\}$. Define $H_t = \bigcap_{i \in X} B_{i,t} \bigcap_{1 \le m \le M} E_{m,t}$. Firstly we could estimate the upper bound of $E_\pi[U(\mathbf{S}^*, \mathbf{v}, \mathbf{R}, q(m^*)) - U(\mathbf{S}^*, \mathbf{v}, \mathbf{R}, q(m_r))]$ by coupling. We can get

$$
\begin{aligned}
&\sum_{r=1}^{T} E_\pi \left[ E[(U(\mathbf{S}_0^r, \mathbf{v}_{r-1}^{UCB}, \mathbf{R}, q_{r-1}^{UCB}(m_r)) - U(\mathbf{S}_0^r, \mathbf{v}, \mathbf{R}, q(m_r)))1(H_{r-1})|\mathcal{F}_{r-1}] \right] \\
\le& C_2 N \sqrt{T \log T} + C_3 M^2 \sqrt{T \log T}.
\end{aligned}
$$

Regarding the second part in Equation (3.2), we now bound the difference between $E_\pi[U(\mathbf{S}^*, \mathbf{v}, \mathbf{R}, q(m_r))]$ and $E_\pi[U(\mathbf{S}^r, \mathbf{v}, \mathbf{R}, q(m_r))]$. Note that $\mathbf{S}^r$ is an updated list that may differ from $\mathbf{S}_0^r$, so we again use coupling to bound the difference. We couple the processes sending $\mathbf{S}^*$ and $\mathbf{S}^r$, which both have the identical dissemination frequency $m_r$.

Summing over all time steps, we have $\sum_{r=1}^{T} E_\pi[U(\mathbf{S}^*, \mathbf{v}, \mathbf{R}, q(m_r)) - U(\mathbf{S}^r, \mathbf{v}, \mathbf{R}, q(m_r))] \le C_4 \sqrt{\log T} \sum_{i=1}^{N} E_\pi \left[ \sum_{t=1}^{T} z_{i,t} \sqrt{\frac{1}{T_i(t-1)}} \right] + D E_\pi \left[ \sum_{t=1}^{T} 1(J_t^c) \right]$ where $J_t = \bigcap_{i \in X} B_{i,t}$. With some derivations, for each $i \in X$, we can bound the term

$$
E_\pi \left[ \sum_{t=1}^{T} z_{i,t} \sqrt{\frac{1}{T_i(t-1)}} \right] \le C_5 \tau_{max} + C_6 \sqrt{T}.
$$

Applying Lemma 3.1, we have

$$
E_\pi \left[ \sum_{t=1}^{T} 1(H_t^c) \right] \le C_7(N + M).
$$

Combining all the results above, we arrive at the desired result. $\qquad\square$

The complete proofs of Lemma 3.2, Lemma 3.3, and Theorem 3.4 are included in the supplementary material. If the delay effect is absent, we can show that the regret bound in Theorem 3.4 is reduced to $C_1(N + M^2)\sqrt{T \log T}$. In other words, we show that the increase in the regret contributed by the delayed feedback is additive and linear in $\tau_{max}$ in our setting. In this paper, we mainly focus on the instance-independent regret, while it is possible to derive an instance-dependent result similarly following Theorem 2 and Theorem 4 in [13], combined with the technique that we have used here to deal with multiple interactions and the delayed feedback.

As a special case of our proposed model, we can compare this result with the state-of-the-art results of cascading bandits. [15] has derived lower bound as $\Omega(\sqrt{NMT})$. Moreover, [11] has shown the regret of $O(\sqrt{NT \log T} + NE[\tau])$. Our setup is more complicated since there are more unknown parameters that need to be learned (probability of abandonment), and the feedback is received in a delayed manner. The regret bound for our proposed algorithm is $O(\max(N, M^2)\sqrt{T} + N\tau)$. From the comparison to two special cases, we can see that regret is optimal in $T$ as well as the delay duration.

## 4 Personalized Recommendation

In the previous section, we assume that all users share the identical preferences towards content and the abandonment behavior. To provide personalized recommendations which incorporate content and

user features, we propose a contextual variant of the learning algorithm which is motivated by [17]. Note that in this section, we allow the message set $X$ to vary for different users, denoted as $X_r$.

Define $\mathbf{w}_{r,i}$ as the features of message $i$ in $X_r$, and $\mathbf{x}_r$ as user's features. We assume the attraction probability of message $i$ is a logit function of its content feature $\mathbf{w}_{r,i}$, and user $r$'s abandonment behavior can be described by a logit function of the user's features $\mathbf{x}_r$. Define the link function $\mu(y) = \exp(y)/(1 + \exp(y))$. That is,

$$1 - P(\text{abandon}|\mathbf{x}_r, m) = \exp(\mathbf{x}_r'\alpha_m)/(1 + \exp(\mathbf{x}_r'\alpha_m)) = \mu(\mathbf{x}_r'\alpha_m), \text{ and}$$

$$P(\text{click on message } i|\mathbf{w}_{r,i}) = \exp(\mathbf{w}_{r,i}'\beta)/(1 + \exp(\mathbf{w}_{r,i}'\beta)) = \mu(\mathbf{w}_{r,i}'\beta).$$

Set $\hat{Y}_{r,j} = 1$ if user $r$ remains in the system after she does not click on the $j^{th}$ message in a list, while $\hat{Y}_{r,j} = 0$ if she abandons the system after viewing the $j^{th}$ message. Therefore, the maximum likelihood estimator of $\alpha_m$ at time $t$, $\hat{\alpha}_{m,t}$, can be obtained as

$$\hat{\alpha}_{m,t} = \arg\max_{\alpha} \sum_{r=1}^{t} \sum_{j=1}^{m_r} 1(m_r = m)\left(\hat{Y}_{r,j}\mathbf{x}_r'\alpha - \log(1 + e^{\mathbf{x}_r'\alpha})\right)$$

$$1(\text{user } r \text{ examines the } j^{th} \text{ message but does not click by time } t). \qquad (4.1)$$

Similarly, let $Y_{r,j} = 1$ if user $r$ clicks on the message $j$, and $Y_{r,i} = 0$ otherwise. The maximum likelihood estimator of $\beta$ at time $t$ can be obtained as

$$\hat{\beta}_t = \arg\max_{\beta} \sum_{r=1}^{t} \sum_{j=1}^{m_r} \left(Y_{r,j}\mathbf{w}_{r,\kappa_r(j)}'\beta - \log\left(1 + e^{\mathbf{w}_{r,\kappa_r(j)}'\beta}\right)\right)$$

$$1(\text{user } r \text{ gives the feedback for the } j^{th} \text{ message by time } t), \qquad (4.2)$$

where feedback includes both clicks and no-clicks and $\kappa_r$ is the index function for user $r$.

Define the covariance matrices at time $t$ for two estimators as $M_{m,t} = \sum_{r=1}^{t} \sum_{j=1}^{m_r} \mathbf{x}_r\mathbf{x}_r'1(\text{user } r \text{ examines the } j^{th} \text{ message but does no click by time } t)1(m_r = m)$, and $V_t = \sum_{r=1}^{t} \sum_{j=1}^{m_r} \mathbf{w}_{r,\kappa(j)}\mathbf{w}_{r,\kappa(j)}'1(\text{user } r \text{ gives the feedback for the } j^{th} \text{ message by time } t)$. We propose the following contextual learning algorithm. In each step, after observing users' feedback, we update the maximum likelihood estimators for $\alpha_m$ and $\beta$, respectively, followed by their upper confidence bound. We then use these estimators to determine the messages for the active users.

---

**Algorithm 3:** An online algorithm for contextual cascading bandits with delayed feedback

---

1 **Input:** Total time steps $T$, tuning parameter $\eta$, $\gamma_1$ and $\gamma_2$; $t = 0$ ;
2 **Initialize:** Randomly choose frequency and messages to send until time $\eta$. Update $V_\eta$ and $M_{m,\eta}$;
3 **while** $t < T$ **do**
4     Update $\hat{\alpha}_{m,t}$ and $\hat{\beta}_t$ by Equation (4.1) and Equation (4.2); $t = t + 1$;
5     Observe a user's contexts $\mathbf{x}_t$ and message contexts $\mathbf{w}_{t,i}$; Update

$$v_{t-1}^{UCB}(\mathbf{w}_{t,i}) = \mu\left(\mathbf{w}_{t,i}'\hat{\beta}_{t-1} + \gamma_1\|\mathbf{w}_{t,i}\|_{V_{t-1}^{-1}}\right), \text{ for all } i, \text{ and}$$

$$q_{m,t-1}^{UCB}(\mathbf{x}_t) = \mu\left(\mathbf{x}_t'\hat{\alpha}_{m,t-1} + \gamma_2\|\mathbf{x}_t\|_{M_{m,t-1}^{-1}}\right), \text{ for all } m$$

    **for** *Any active user $r$ with message scheduled to be sent at time $t$* **do**
6         Compute $\mathbf{S}^r = \arg\max_{\mathbf{S}} E[U(\mathbf{S}, \mathbf{v}_{t-1}^{UCB}, \mathbf{R}, q_{t-1}^{UCB}(m)]$ according to Algorithm 1;
7         Send the first message $e$ in $\mathbf{S}^r \backslash O^r$ to user $r$; $O_r = O_r \cup \{e\}$;
8     **end**
9     Compute $(\mathbf{S}^t, m_t) = \arg\max_{\mathbf{S},m} E[U(\mathbf{S}, \mathbf{v}_{t-1}^{UCB}, \mathbf{R}, q_{t-1}^{UCB}(m)]$ according to Algorithm 1 ;
10     Offer $S_1^t$ to user $t$ and the dissemination frequency is set to $f(m_t)$; $O_t = S_1^t$;
11     Update $M_{m,t}$ and $V_t$;
12 **end**

---

It is also important to note that although Algorithm 2 and Algorithm 3 are both based on the assumption that messages cannot be recommended repeatedly, they can be easily generalized. If

$O_r$ is not excluded from the candidate messages, the two algorithms can be applied the repeat-recommendation case. Moreover, if we want to constrain the total number of repetitive displays of one message as $B$, we can create $B$ copies of that message in the candidate set.

## 5   Numerical Experiment

In this section, we evaluate the performance of both the non-contextual and contextual algorithms based on the features of the real AmEx User Click dataset[3], which records over 463,000 recommendations of AmEx from July 2 to July 7 in 2017. The combination of "product"and "campaign_id" is regarded as messages in simulation.

**Experiment I: Non-contextual setting.** There are $N = 25$ available messages. The attraction probability $v$ is uniformly generated from distribution $[0, 0.5]$, and the return $R$ uniformly generated from distribution $[1, 3]$, The maximum length of message list $M$ is set to be 10. Based on estimations from the short lists in AmEx dataset, we set $q(m) = \frac{1}{1+\exp(0.03m)}$. We set the re-targeting window $D = 200$ in all settings. The response time $\tau$ is uniformly generated from $[0, 3]$ for each user. In addition to our proposed algorithm, we introduce several algorithms as benchmarks: three different $\epsilon$-greedy algorithms and TS-Cascade, which is a Thompson sampling algorithm for cascading bandits with Gaussian update [25].

**Result:** We observe the results of 100 independent simulations for $T = 30,000$ time steps. The results are shown in Fig 2, where the shaded area represents the $95\%$ confidence region. We find that the optimistic greedy and decaying $\epsilon$-greedy algorithms are difficult to find an optimal list. The average regret at $T = 30,000$ of our UCB algorithm is $550.14$, with a $95\%$ confidence interval of $(454.71, 645.57)$, significantly better than the benchmarks.

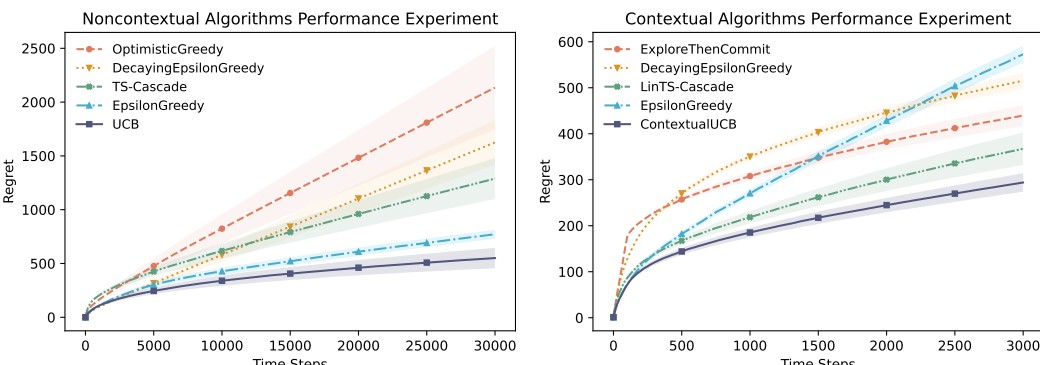

Figure 2: Regrets of Algorithm 2.                     Figure 3: Regrets of Algorithm 3.

**Experiment II: Contextual setting.** There are $N = 25$ available messages. The maximum length of message list $M$ is set to be 20. User features are uniformly generated from $[0, 5] \times [0, 5] \times [0, 5]$, and message features are uniformly generated from $[-6, 0] \times [0, 1] \times [0, 2] \times [-5, 0]$. The coefficient related to the abandonment behavior, denoted as $\alpha_m$, is uniformly generated in the range (four-dimensional including the intercept) $-1.04 \times [-0.064m, 0] \times [-0.08m, 0] \times [-0.16m, 0]$ for $m = 1, \cdots, 20$, where $\alpha_{m,1}$ is the intercept. An alternative coefficient $\tilde{\alpha}_m$ is uniformly generated from $-1.04 \times [-0.004m, 0] \times [-0.064m, 0] \times [-0.08m, 0]$ for $m = 1, \cdots, 20$. The coefficient related to message attraction is $\beta = (0.05, 0.2, 0.1, 0.3, 0.4)$, where $\beta_1$ is the intercept. Users' response time is uniformly distributed on [0,10]. We also implement several benchmark algorithms: two variants of the $\epsilon$-greedy algorithm, the explore-then-commit algorithm, and LinTS-Cascade, which is a linear generalization of TS-Cascade [25].

**Result:** We observe the results of 50 independent simulations for $T = 3,000$ time steps. The result is shown in Fig 3. The average regret at $T = 3,000$ of Algorithm 3 is $293.94$, with a $95\%$ confidence interval of $(273.85, 314.04)$, significantly better than the benchmark algorithms. In the early stage,

---

[3]https://www.kaggle.com/code/muditagrawal/amex-user-click-prediction

the contextual UCB algorithm also has good performance, showing a faster convergence speed than benchmarks.

**Experiment III: Comparison of UCB and contextual UCB.** To compare the effect of Algorithm 2 and Algorithm 3, we make two settings: (a): $N = 25, M = 20$ with user features and message features generated from the same distribution as in Experiment 2, but only generated once and then fixed. All other parameters are the same as in Experiment 2; (b): same as (a) but with $N = 100$.

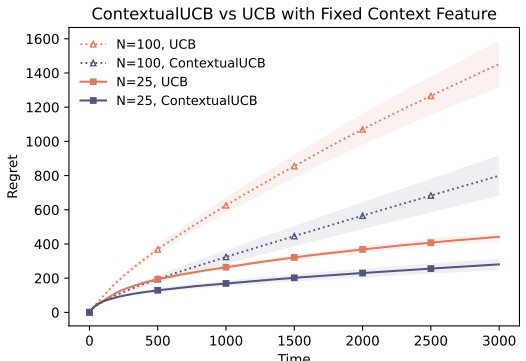

Figure 4: Comparison of the two UCB algorithms.

**Result:** We observe the results of 50 independent simulations for $T = 3,000$ time steps. The result is shown in Fig 4. The average regrets under the 4 settings are $1451.95, 798.94, 441.90$ and $280.73$ from high to low, respectively. The contextual bandit UCB always work better than UCB algorithm in the same setting. Furthermore, the advantage of contextual algorithm becomes greater when there are more messages. This is because more messages will improve the estimation accuracy of $\alpha$ and $\beta$.

## 6  Conclusion

In this work, we considered a cascading bandit problem with delayed feedback and flexible rewards. Based on the users' feedback, the learning agent needs a policy to select a list of messages and determine the frequency at which these messages are disseminated. We proposed a novel learning algorithm to effectively learn such a policy, and it is capable of recommending a list that is dynamic to a given user within polynomial time. Experiments proved the effectiveness and applicability of our algorithm compared to benchmark algorithms in a novel cascading bandit problem.

For future work, it would be interesting to analyze the performance of alternative bandit algorithms, such as Thompson-sampling-based algorithms for this setting. Additionally, considering that users' preferences may change over time, another avenue for future research would be to incorporate non-stationarity into user preferences.

## Acknowledgments and Disclosure of Funding

Dairui Wang, Yan Zhang, and Wei Qi acknowledge the support from National Natural Science Foundation of China (Grants 72242106, 72188101).

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
