# Cascading Bandits: Optimizing Recommendation Frequency in Delayed Feedback Environments
## Supplementary Material

## 1   Notations

- $\kappa$ : index function, $\kappa(i) = j$ if and only if $S_i = \{j\}$

- $\vartheta$ : inverse function of $\kappa$, i.e., $\vartheta(i) = j$ if and only if $S_j = \{i\}$

- $\mathbf{v} = (v_1, v_2, \cdots, v_N)$: attraction probabilities of messages

- $\mathbf{R} = (R_1, R_2, \cdots, R_N)$: reward of messages

- $D$: re-targeting window

- $f(m) = \lceil m/D \rceil$: frequency when the total number of messages is $m$

- $q(m)$ : the probability of staying in the system after skipping a message (i.e., no click) for users with dissemination frequency $f(m)$

- $\mathbf{q} = (q(1), q(2), \cdots, q(M))$: a vector of $q$ functions

- $w_i(m)$: the examine probability of message $i$ when the total number of messages is $m$

- $U(\mathbf{S}, \mathbf{v}, q)$: the total payoff for sequence $\mathbf{S}$ when parameters are $\mathbf{v}$ and $q$

- $\gamma_i$: the characteristic parameter of message $i$, $\gamma_i = \frac{v_i R_i}{1 - v_i(1 - q(m))}$

- $T_i(t)$: the total number of feedback (i.e., sum of clicks and no-clicks) received for message $i$ by time $t$

- $c_i(t)$: the number of clicks for message $i$ by time $t$

- $\tilde{T}_m(t)$: the total number of no-clicks from users with dissemination frequency $f(m)$

- $b_m(t)$: the number of abandoned users with frequency $f(m)$ by time $t$

- $n_m(t)$: equals $\tilde{T}_m(t) - b_m(t)$

- $\mathcal{E}_r : t \in \mathcal{E}_r$ if the agent sends message to user $r$ at time $t$

- $\epsilon_m(t)$ : the set of time stamps that a message with frequency $f(m)$ is sent

- $\rho_r^k$: the time stamps when the $k^{th}$ message is sent to user $r$

- $m_r$ : the number of total messages for user $r$ and the corresponding frequency is $f(m_r)$

- $e_{t,k}^r$ : the index of the $k^{th}$ message sent to user $r$ at time $t$

- $O_r^t$: the messages which have been sent to user $r$ by time $t$

- $z_{i,t}$: the total number of times that message $i$ is sent to users at time $t$

- $A_i(t)$: the set of time stamps of sending message $i$ by time $t$

- $\mathbf{w}_{r,i}$: the features of message $i$ at time $r$

- $\mathbf{x}_r$ : the features of user $r$

- $\alpha_m$ : coefficients related to abandonment behavior when the frequency is $f(m)$

- $\beta$: coefficients related to the attraction probability of messages

- $Y_{r,i}$: $Y_{r,i} = 1$ if user $r$ clicks on the message $i$, and $Y_{r,i} = 0$ otherwise

- $\hat{Y}_{r,j}$ : $\hat{Y}_{r,j} = 1$ if user $r$ remains in the system after she does not click on the $j^{th}$ message in a list, while $\hat{Y}_{r,j} = 0$ otherwise

## 2    Proofs

Throughout the paper, we will use coupling to prove several key results. For more information on this, we refer the reader to Section 2.2 in [1].

**Theorem 2.2** *In the optimal sequence $\mathbf{S}^*$, the characteristic parameter of messages $\gamma = \frac{vR}{1-v(1-q(m))}$ are sorted in a descending order.*

**Proof.** We prove this theorem by contradiction. Assume the optimal sequence

$$\mathbf{S}^* = (S_1, S_2, \cdots, S_i, S_{i+1}, \cdots, S_m),$$

with $\gamma_{\kappa(i)} < \gamma_{\kappa(i+1)}$, which implies $v_{\kappa(i)}R_{\kappa(i)}(1 - v_{\kappa(i+1)}(1 - q(m))) < v_{\kappa(i+1)}R_{\kappa(i+1)}(1 - v_{\kappa(i)}(1 - q(m)))$. The expected reward

$$
\begin{aligned}
&E[U(\mathbf{S}^*, \mathbf{v}, \mathbf{R}, q(m))] \\
&= \sum_{k=1}^{m} [v_{\kappa(k)} R_{\kappa(k)} \prod_{s=1}^{k-1} q(m)(1 - v_{\kappa(s)})] \\
&= \sum_{1 \leq k \leq m, k \neq i, i+1} [v_{\kappa(k)} R_{\kappa(k)} \prod_{s=1}^{k-1} q(m)(1 - v_{\kappa(s)})] + v_{\kappa(i)} R_{\kappa(i)} \prod_{s=1}^{i-1} q(m)(1 - v_{\kappa(s)}) \\
&\quad + v_{\kappa(i+1)} R_{\kappa(i+1)} \prod_{s=1}^{i-1} q(m)(1 - v_{\kappa(s)})((1 - v_{\kappa(i)})q(m)).
\end{aligned}
$$

Consider the sequence $\mathbf{S}' = (S_1, S_2, \cdots, S_{i+1}, S_i, \cdots, S_m)$. Similarly we have

$$E[U(\mathbf{S}', \mathbf{v}, \mathbf{R}, q(m))]$$

$$= \sum_{1 \le k \le m, k \ne i, i+1} [v_{\kappa(k)} R_{\kappa(k)} \prod_{s=1}^{k-1} q(m)(1 - v_{\kappa(s)})] + v_{\kappa(i+1)} R_{\kappa(i+1)} \prod_{s=1}^{i-1} q(m)(1 - v_{\kappa(s)})$$

$$+ v_{\kappa(i)} R_{\kappa(i)} \prod_{s=1}^{i-1} q(m)(1 - v_{\kappa(s)})((1 - v_{\kappa(i+1)})q(m)).$$

Thus,

$$E[U(\mathbf{S}^*, \mathbf{v}, \mathbf{R}, q(m))] - E[U(\mathbf{S}', \mathbf{v}, \mathbf{R}, q(m))]$$

$$= \prod_{s=1}^{i-1} ((1 - v_{\kappa(s)})q(m))$$

$$\cdot [v_{\kappa(i)} R_{\kappa(i)} + v_{\kappa(i+1)} R_{\kappa(i+1)}((1 - v_{\kappa(i)})q(m)) - v_{\kappa(i+1)} R_{\kappa(i+1)} - v_{\kappa(i)} R_{\kappa(i)}((1 - v_{\kappa(i)})q(m))]$$

$$< 0.$$

It contradicts with the assumption that $\mathbf{S}^*$ is the optimal sequence. Therefore, the characteristic parameter of messages $\gamma = \frac{vR}{1 - v(1 - q(m))}$ are sorted in a descending order. $\square$

**Lemma 3.1** *For any $t$, we have $P\left(v_{i,t}^{UCB} - \sqrt{8\frac{\log t}{T_i(t)}} < v_i < v_{i,t}^{UCB}\right) \ge 1 - \frac{2}{t^4}$ for all $i \in X$ and $P\left(q_t^{UCB}(m) - \sqrt{8\frac{\log t}{\tilde{T}_m(t)}} < q(m) < q_t^{UCB}(m)\right) \ge 1 - \frac{2}{t^4}$ for all $1 \le m \le M$.*

**Proof.** Firstly, it is easy to verify that $\hat{v}_{i,t}$ and $\hat{q}_t(m)$ are unbiased estimators. Applying Hoeffding's inequality, we have

$$P\left(v_{i,t}^{UCB} < v_i\right) + P\left(v_{i,t}^{UCB} > v_i + 2\sqrt{2 \log t / T_i(t)}\right)$$

$$= P\left(\hat{v}_{i,t} + \sqrt{2 \log t / T_i(t)} < v_i\right) + P\left(\hat{v}_{i,t} > v_i + \sqrt{2 \log t / T_i(t)}\right)$$

$$= P\left(|\hat{v}_{i,t} - v_i| > \sqrt{2 \log t / T_i(t)}\right) \le 2 \exp(-4 \log t) = \frac{2}{t^4}.$$

It implies that

$$P\left(v_{i,t}^{UCB} - \sqrt{\frac{8 \log t}{T_i(t)}} < v_i < v_{i,t}^{UCB}\right) \ge 1 - \frac{2}{t^4}.$$

Similarly, we have

$$P\left(q_t^{UCB}(m) < q(m)\right) + P\left(q_t^{UCB}(m) > q(m) + 2\sqrt{2 \log t / \tilde{T}_m(t)}\right)$$

$$= P\left(\hat{q}_t(m) + \sqrt{2 \log t / \tilde{T}_m(t)} < q(m)\right) + P\left(\hat{q}_t(m) > q(m) + \sqrt{2 \log t / \tilde{T}_m(t)}\right)$$

$$= P\left(|\hat{q}_t(m) - q(m)| > \sqrt{2 \log t / \tilde{T}_m(t)}\right) \le 2 \exp(-4 \log(t)) = \frac{2}{t^4},$$

which implies that

$$P\left(q_t^{UCB}(m) - \sqrt{\frac{8\log t}{\tilde{T}_m(t)}} < q(m) < q_t^{UCB}(m)\right) \geq 1 - \frac{2}{t^4}.$$

$\square$

**Lemma 3.2** *Assume $\mathbf{S}^*$ is the optimal sequence of messages with corresponding total message $m^*$. Under the condition that $0 \leq \mathbf{v} \leq \mathbf{v}^{UCB}$ and $0 \leq q(m^*) \leq q^{UCB}(m^*)$, we have*

$$E[U(\mathbf{S}^*, \mathbf{v}^{UCB}, \mathbf{R}, q^{UCB}(m^*))] \geq E[U(\mathbf{S}^*, \mathbf{v}, \mathbf{R}, q(m^*))].$$

**Proof.** This Lemma describes the monotonic increasing property of total payoff with respect to both $\mathbf{v}$ and $q(m^*)$. First, we couple the recommending process of $(\mathbf{S}^*, \mathbf{v}, \mathbf{R}, q^{UCB}(m^*))$ (call this process targets on user 1) and $(\mathbf{S}^*, \mathbf{v}, \mathbf{R}, q(m^*))$ (call this process targets on user 2). Generate $M$ independent random variables $u_j$ for $j = 1, \ldots, M$ which all follow the uniform distribution on [0,1]. The event $u_j < q(m^*)$ means that both users will stay after observing the $j^{th}$ unsatisfying message, while the event $u_j > q^{UCB}(m^*)$ means that both users will leave. $q(m^*) \leq u_j \leq q^{UCB}(m^*)$ means that user 1 will stay and user 2 will leave, in which case the coupling breaks. In all cases, recommending the sequence $\mathbf{S}^*$ with parameters $(\mathbf{v}, q^{UCB}(m^*))$ will have at least the same payoff as recommending the sequence $\mathbf{S}$ with parameter $(\mathbf{v}, q(m^*))$. Therefore, the increasing property of total payoff with respect to $q(m^*)$ has been proven.

Then consider two identical recommending lists with the attraction probability of only one message is different. Assume the $k$-th message has $v_{k1} > v_{k2}$, and $v_{i1} = v_{i2}, \quad \forall i \neq k$. Denote the expected return of the two lists as $E[U_1]$ and $E[U_2]$, respectively. We have

$$\begin{aligned}
&E[U_1] - E[U_2] \\
=&(1 - v_{bef})(v_{k1}R_k + (1 - v_{k1})R_{after} - v_{k2}R_k - (1 - v_{k2})R_{after}) \\
=&(1 - v_{bef})(v_{k1} - v_{k2})(R_k - R_{after}) \\
\geq&0,
\end{aligned}$$

with $R_{bef}, R_{after}$ means the expected return of all the messages before/after the $k$-th message respectively. The last inequality holds because $R_k \geq R_{after}$, otherwise removing the $k$-th message will give a higher return $R_{after}$. Since $k$ can be any message, we have proven the increasing property of total payoff with respect to $\mathbf{v}$. $\square$

**Lemma 3.3** *When all messages have identical reward, for $t \in \mathcal{E}_r$ and any $q' \in (0, 1)$, we have*

$$E_\pi[E[(U(k, \mathbf{v}, \mathbf{R}, q') - U(e_{t,k}^r, \mathbf{v}, \mathbf{R}, q'))1(\mathbf{v}_t^{UCB} \geq \mathbf{v})|\mathcal{F}_{t-1}]]$$
$$\leq E_\pi\left[E\left[\left(v_{e_{t,k}^r, t}^{UCB} - v_{e_{t,k}^r}\right)1(\mathbf{v}_t^{UCB} \geq \mathbf{v})|\mathcal{F}_{t-1}\right]\right],$$

*where $e_{t,k}^r$ is the index of the $k^{th}$ message sent to user $r$ at time $t$.*

**Proof.** Given the assumption, $v_1 \geq v_2 \geq \cdots \geq v_N$ in the optimal list. If $e^r_{t,k} \leq k$, the conclusion holds because $v_{e^r_{t,k}} \geq v_k$, which implies that $E[U(e^r_{t,k}, \mathbf{v}, \mathbf{R}, q)] \geq E[U(k, \mathbf{v}, \mathbf{R}, q)]$. Otherwise, if $e^r_{t,k} > k$, then $v_{e^r_{t,k}} \leq v_k$. Note that $v^{UCB}_{e^r_{t,k},t}$ is at least the $k^{th}$ largest among $\mathbf{v}^{UCB}_t$, otherwise $e^r_{t,k}$ will not be chosen. With $\mathbf{v}^{UCB}_t \geq \mathbf{v}$, we have $v_k \leq v^{UCB}_{e^r_{t,k},t}$ because the $k^{th}$ largest value in sequence $\mathbf{v}^{UCB}_t$ is larger than or equal to the $k^{th}$ largest value in $\mathbf{v}$. Therefore, we have $v_{e^r_{t,k}} \leq v_k \leq v^{UCB}_{e^r_{t,k},t}$. It implies that $v_k - v_{e^r_{t,k}} \leq v^{UCB}_{e^r_{t,k},t} - v_{e^r_{t,k}}$. Thus, we have reached the desired result. □

**Theorem 3.4** *The expected regret of Algorithm 2 is bounded above by*

$$Reg(T) \leq C_1(N + M^2)\sqrt{T \log T} + C_2 N \tau_{max}$$

*for some constants $C_1$ and $C_2$.*

**Proof.** Firstly, we show that $Reg(T) \leq CReg_{iden}(T)$, where $Reg_{iden}(T)$ denotes the regret of Algorithm 2 with messages with identical reward. Define $R_{max}$ to be the maximum in the actual list, and $S^*$ to be the optimal list. Thus, $Reg(T) \leq R_{max} U^*_{max} \leq R_{max} C'(U^* - U^S) = CReg_{iden}(T)$. The second inequality holds because $U^*_{max}$ can be bounded above and $C'$ can always be fixed selected in a specific problem.

Then we only need to discuss the identified-reward scenario. We omit the notation $\mathbf{R}$ in the proof below within this scenario. Define the optimal length of message is $m^*$ with the corresponding optimal staying probability $q_* = q(m^*)$. Assume the sequence offered to user $r$ (entering at time $r$) is $\mathbf{S}^r$ with total message number $m_r$. We want to quantify the difference between the expected profit gained from $\mathbf{S}^r$ and $\mathbf{S}^*$ where $\mathbf{S}^* = (1, 2, \cdots, m^*)$. First we note that

$$\begin{aligned}
&E_\pi[U(\mathbf{S}^*, \mathbf{v}, q_*)] - E_\pi[U(\mathbf{S}^r, \mathbf{v}, q(m_r))] \\
&= E_\pi[U(\mathbf{S}^*, \mathbf{v}, q_*)] - E_\pi[U(\mathbf{S}^*, \mathbf{v}, q(m_r))] + E_\pi[U(\mathbf{S}^*, \mathbf{v}, q(m_r))] - E_\pi[U(\mathbf{S}^r, \mathbf{v}, q(m_r))].
\end{aligned} \quad (1)$$

Let $\mathbf{S}^r_0$ denote the recommendation sequence for user $r$ when she enters the system, i.e., $\mathbf{S}^r_0$ is the optimal sequence given $\mathbf{v}^{UCB}_{r-1}$ and $\mathbf{q}^{UCB}_{r-1}$. Note that this list may change at a later time when more information becomes available. Define events

$$B_{i,t} = \left\{ v^{UCB}_{i,t} - \sqrt{8 \frac{\log t}{T_i(t)}} < v_i < v^{UCB}_{i,t} \right\} \text{ and } E_{m,t} = \left\{ q^{UCB}_t(m) - \sqrt{8 \frac{\log t}{\hat{T}_m(t)}} < q(m) < q^{UCB}_t(m) \right\}.$$

Define $H_t = \bigcap_{i \in X} B_{i,t} \bigcap_{1 \leq m \leq M} E_{m,t}$ and $J_t = \bigcap_{i \in X} B_{i,t}$. On event $H_t$, firstly we have

$$\begin{aligned}
E_\pi[U(\mathbf{S}^r_0, \mathbf{v}, q(m_r))] &\leq E_\pi[U(\mathbf{S}^*, \mathbf{v}, q(m_r))] \leq E_\pi[R(\mathbf{S}^*, \mathbf{v}, q(m^*))] \\
&\leq E_\pi[U(\mathbf{S}^*, \mathbf{v}^{UCB}_{r-1}, q^{UCB}_{r-1}(m^*))] \leq E_\pi[U(\mathbf{S}^r_0, \mathbf{v}^{UCB}_{r-1}, q^{UCB}_{r-1}(m_r))],
\end{aligned}$$

where the first inequality holds because $\mathbf{S}^*$ is the optimal order (arranged from the highest attraction probability to the lowest), the second inequality holds because $q(m^*)$ is the staying probability corresponding to the optimal frequency $m^*$, the third inequality holds because of Lemma 3.2, and the fourth inequality holds because $\mathbf{S}^r_0$ is the optimal sequence given values $\mathbf{v}^{UCB}_{r-1}$ and $\mathbf{q}^{UCB}_{r-1}$. Thus we have

$$E_\pi[(U(\mathbf{S}^*, \mathbf{v}, q(m^*)) - U(\mathbf{S}^r, \mathbf{v}, q(m_r)))1(H_{r-1})]$$

$$\leq E_\pi[(U(\mathbf{S}_0^r, \mathbf{v}_{r-1}^{UCB}, q_{r-1}^{UCB}(m_r)) - U(\mathbf{S}_0^r, \mathbf{v}, q(m_r)))1(H_{r-1})].$$

To get the difference between the expected payoff of two items above, we use coupling to bound the difference between the recommending process $\mathbf{S}_0^r$ with $q_{r-1}^{UCB}(m_r), \mathbf{v}_{r-1}^{UCB}$ (call this process targets on user 1) and $\mathbf{S}_0^r$ with $q(m_r), \mathbf{v}$ (call this process targets on user 2). For the $k^{th}$ recommendation where $k$ ranges from 1 to $m_r$, generate two independent uniform random variables $w_1 \sim unif[0,1]$ and $w_2 \sim unif[0,1]$. The event $w_1 \leq v_{\mathbf{S}_0^r(k)}$ means that both click on the $k^{th}$ message. The event $w_1 \geq v_{\mathbf{S}_0^r(k),r-1}^{UCB}$ means that both do not click on the $k^{th}$ message. If $v_{\mathbf{S}_0^r(k)} \leq w_1 \leq v_{\mathbf{S}_0^r(k),r-1}^{UCB}$, the coupling process breaks, i.e., user 1 clicks on the $k^{th}$ message but user 2 does not click on the message. The event $w_2 \leq q(m_r)$ denotes that both stay in the system. If $w_2 \geq q_{r-1}^{UCB}(m_r)$, both exit the system. If $q(m_r) < w_2 < q_{r-1}^{UCB}(m_r)$ and $w_1 \geq v_{\mathbf{S}_0^r(k),r-1}^{UCB}$, user 1 chooses to stay in the system and user 2 exits the system, so the coupling process breaks. Let $\hat{\tau}_r$ denote the stopping time that the coupling process breaks. Also define $\varepsilon_m$ as the set of time stamps that a message with frequency $f(m)$ is sent to user and $\rho_r^k$ as the time to offer the $k^{th}$ message to user $r$. Thus we have

$$E_\pi\left[E[(U(\mathbf{S}_0^r, \mathbf{v}_{r-1}^{UCB}, q_{r-1}^{UCB}(m_r)) - U(\mathbf{S}_0^r, \mathbf{v}, q(m_r)))1(H_{r-1})|\mathcal{F}_{r-1}]\right]$$

$$\leq E_\pi\left[E\left[\sum_{k=1}^{m_r} 1(\hat{\tau}_r = k)1(H_{r-1})|\mathcal{F}_{r-1}\right]\right]$$

$$\leq E_\pi\left[E\left[\sum_{k=1}^{m_r}\sum_{i=1}^{N} 1(i \in \mathbf{S}_0^r(k))\left(v_{i,r-1}^{UCB} - v_i\right)1(H_{r-1})|\mathcal{F}_{r-1}\right]\right]$$

$$+ E_\pi\left[E\left[\sum_{k=1}^{m_r} 1(\rho_r^k \in \varepsilon_{m_r})(q_{r-1}^{UCB}(m_r) - q(m_r))1(H_{r-1})|\mathcal{F}_{r-1}\right]\right]$$

$$\leq E_\pi\left[\sum_{k=1}^{m_r}\sum_{i=1}^{N} 1(i \in \mathbf{S}_0^r(k))\sqrt{8\frac{\log(r-1)}{T_i(r-1)}}\right] + E_\pi\left[\sum_{k=1}^{m_r} 1(\rho_r^k \in \varepsilon_{m_r})\sqrt{8\frac{\log(r-1)}{n_{m_r}(r-1)}}\right].$$

Summing over all the time steps, we have

$$\sum_{r=1}^{T} E_\pi\left[E[(U(\mathbf{S}_0^r, \mathbf{v}_{r-1}^{UCB}, q_{r-1}^{UCB}(m_r)) - U(\mathbf{S}_0^r, \mathbf{v}, q(m_r)))1(H_{r-1})|\mathcal{F}_{r-1}]\right]$$

$$\leq C_1\sqrt{\log T}\sum_{r=1}^{T} E_\pi\left[\sum_{i=1}^{N}\sum_{k=1}^{m_r} 1(i \in \mathbf{S}_0^r(k))\sqrt{\frac{1}{T_i(r-1)}}\right]$$

$$+ C_1\sqrt{\log T}\sum_{r=1}^{T} E_\pi\left[\sum_{k=1}^{m_r} 1(\rho_r^k \in \varepsilon_{m_r})\sqrt{\frac{1}{n_{m_r}(\rho_r^k - 1)}}\right]$$

$$\leq C_2\sqrt{\log T}\sum_{i=1}^{N} E_\pi[\sqrt{T_i(T)}] + C_2 M\sqrt{\log T}\sum_{m=1}^{M}\sum_{t=1}^{T} E_\pi\left[1(t \in \varepsilon_m)\sqrt{\frac{1}{n_m(t-1)}}\right].$$

If $t \in \varepsilon_m$, then the user has at least probability $1 - v_{max}$ to reject the message, in which case the user has the choice to abandon the system. Therefore, if $t \in \varepsilon_m$, $n_m(t+1) = n_m(t) + 1$ with

probability at least $1 - v_{max}$. It implies that for any $m = 1 \cdots M$,

$$\sum_{t=1}^{T} E_\pi \left[ 1(t \in \varepsilon_m) \sqrt{\frac{1}{n_m(t-1)}} \right] \leq \frac{1}{1 - v_{max}} E_\pi[\sqrt{n_m(T)}] \leq C_3 E_\pi[\sqrt{n_m(T)}].$$

Since $\sum_{m=1}^{M} n_m(T) \leq TM$ with probability 1, we have

$$\sum_{m=1}^{M} E_\pi[\sqrt{n_m(T)}] \leq M\sqrt{T}.$$

Since $\sum_{i=1}^{N} T_i(T) \leq \min(M, N)T$ with probability 1, we have

$$\sum_{i=1}^{N} E_\pi[\sqrt{T_i(T)}] \leq \sqrt{N \min(M, N)T}.$$

Thus, we get the inequality that

$$\sum_{r=1}^{T} E_\pi \left[ E[(U(\mathbf{S}_0^r, \mathbf{v}_{r-1}^{UCB}, q_{r-1}^{UCB}(m_r)) - U(\mathbf{S}_0^r, \mathbf{v}, q(m_r)))1(H_{r-1})|\mathcal{F}_{r-1}] \right]$$
$$\leq C_2 N \sqrt{T \log T} + C_3 M^2 \sqrt{T \log T}.$$

Applying Lemma 3.1, we have

$$\sum_{r=1}^{T} E_\pi[1(H_r^c)] \leq \sum_{r=1}^{T} \sum_{i=1}^{N} E_\pi[1(B_{i,r}^c)] + \sum_{r=1}^{T} \sum_{m=1}^{M} E_\pi[1(E_{m,r}^c)]$$
$$\leq N \sum_{t=1}^{T} \frac{2}{t^4} + M \sum_{t=1}^{T} \frac{2}{t^4} \leq C_4(N + M).$$

For Equation (1), now we bound the difference between $E_\pi[U(\mathbf{S}^*, \mathbf{v}, q(m_r))]$ and $E_\pi[U(\mathbf{S}^r, \mathbf{v}, q(m_r))]$. Note that $\mathbf{S}^r$ is an adapted sequence, which can be different from $\mathbf{S}_0^r$, so we use coupling to bound the difference. We couple the recommending process of $\mathbf{S}^*$ (call this to user 1) and $\mathbf{S}^r$ (call this to user 2) when the total number of messages is $m_r$. For the $k^{th}$ recommending message at time $t$ to user $r$, set $a_1 = \min\{v_k, v_{e_{t,k}^r,t}\}$ and $a_2 = \max\{v_k, v_{e_{t,k}^r,t}\}$. Generate two independent uniform random variables $w_1 \sim unif[0,1]$ and $w_2 \sim unif[0,1]$. The event $w_1 < a_1$ denotes that both click on the $k^{th}$ message. If $w_1 \geq a_2$, both do not choose the $k^{th}$ recommending message. When $v_{e_{t,k}^r,t} < v_k$, $a_1 \leq w_2 < a_2$ means that the $k^{th}$ message is chosen in $\mathbf{S}^*$ but not in $\mathbf{S}^r$, and vice versa. Either case means that the coupling process breaks. If $w_2 \geq q(m_r)$, then both exit the system. Otherwise, they will both get the next message unless the whole sequence has run out. Define the stopping time $\tilde{\tau}_r$ as the time that the coupling breaks for user $r$, i.e., the recommendation in $\mathbf{S}^*$ with parameters $\mathbf{v}$ and $q(m_r)$ is a success but that in $\mathbf{S}^r$ with parameters $\mathbf{v}$ and $q(m_r)$ is a failure. Then we have

$$E_\pi[U(\mathbf{S}^*, \mathbf{v}, q(m_r))] - E_\pi[U(\mathbf{S}^r, \mathbf{v}, q(m_r))] \leq E_\pi \left[ \sum_{k=1}^{m_r} 1(\tilde{\tau}_r = k) \right].$$

Now we consider another recommending process $\mathbf{S}^r$ with message value $v_{e^r_{t,k},t}^{UCB}$ where $t = \rho_r^k$ for $k = 1, \cdots, m_r$. Use the same process to couple $\mathbf{S}^r$ with parameter $v_{e^r_{t,k},t}^{UCB}$ and $\mathbf{v}$. Define $\tau_r'$ as the stopping time. On the event that $\mathbf{v}_{\rho_r^k}^{UCB} \geq \mathbf{v}$ for $k = 1, \cdots, m_l$ and $\mathbf{q}_{\rho_r^k}^{UCB} \geq \mathbf{q}$, we have

$$E\left[\sum_{k=1}^{m_r} 1(\tilde{\tau}_r = k)\right] \leq E\left[\sum_{k=1}^{m_r} 1(\tau_r' = k)\right].$$

Recall that $J_t = \bigcap_{i \in X} B_{i,t}$. We therefore have

$$E_\pi\left[E\left[(U(\mathbf{S}^*, \mathbf{v}, q(m_r),) - U(\mathbf{S}^r, \mathbf{v}, q(m_r)))\prod_{k=1}^{m_r} 1(J_{\rho_r^k - 1})\middle| \mathcal{F}_{r-1}\right]\right]$$

$$\leq E_\pi\left[\sum_{k=1}^{m_r} 1(\tau_r' = k)1(J_{\rho_r^k - 1})|\mathcal{F}_{r-1}\right]$$

$$= E_\pi\left[\sum_{k=1}^{m_r}\sum_{i=1}^{N} 1(i \in S_k^r)\left(v_{i,\rho_r^k - 1}^{UCB} - v_i\right) 1(J_{\rho_r^k - 1})\right]$$

$$\leq E_\pi\left[\sum_{k=1}^{m_r}\sum_{i=1}^{N} 1(i \in S_k^r)\sqrt{8\frac{\log(\rho_r^k - 1)}{T_i(\rho_r^k - 1)}}\right].$$

Define $z_{i,t}$ as the total number of times that message $i$ is sent to users at time $t$. If none of item $i$ is recommended at time $t$, $z_{i,t} = 0$. Define $A_i(t)$ as the set of time of recommending $i$ before time $t$. Summing over all users, we have

$$\sum_{r=1}^{T} E_\pi\left[U(\mathbf{S}^*, \mathbf{v}, q(m_r)) - U(\mathbf{S}^r, \mathbf{v}, q(m_r))\right]$$

$$\leq E_\pi\left[\sum_{t=1}^{T}\sum_{i=1}^{N} z_{i,t}\sqrt{8\frac{\log t}{T_i(t-1)}}\right] + \sum_{r=1}^{T} E_\pi\left[\sum_{k=1}^{m_r} 1(J_{\rho_r^k - 1}^c)\right]$$

$$\leq C_5\sqrt{\log T}\sum_{i=1}^{N} E_\pi\left[\sum_{t=1}^{T} z_{i,t}\sqrt{\frac{1}{T_i(t-1)}}\right] + DE_\pi\left[\sum_{t=1}^{T} 1(J_t^c)\right],$$

where $D$ is the duration of the recommending horizon and $C$ is some constant. The last inequality $\sum_{r=1}^{T} E_\pi\left[\sum_{k=1}^{m_r} 1(J_{\rho_r^k - 1}^c)\right] \leq DE_\pi\left[\sum_{t=1}^{T} 1(J_t^c)\right]$ holds because the total recommending duration is at most $D$, which implies that for any $r$ and $k$, $\rho_r^k \leq r + D$. Because of the delayed feedback, user response will be received after at most $\tau_{max}$ time periods. Recall that $\tau$ is the delayed time, so we have

$$T_i(t) \geq \sum_{s \in A_i(t)}\sum_{j=1}^{z_{i,s}} 1(\tau \leq (t - s)).$$

Since we assume $\tau$ has finite support and the maximum possible value is $\tau_{max}$, an obvious bound would be

$$T_i(t) \geq \sum_{s \in A_i(t - \tau_{max})} z_{i,s}.$$

We thus have for each $i \in X$,

$$E_\pi\left[\sum_{t=1}^{T} z_{i,t}\sqrt{\frac{1}{T_i(t-1)}}\right] \le E_\pi\left[\sum_{t=1}^{\tau_{max}} z_{i,t}\right] + E_\pi\left[\sum_{t=\tau_{max}+1}^{T} z_{i,t}\sqrt{\frac{1}{\sum_{s=1}^{t-\tau_{max}} z_{i,s}}}\right]$$

$$\le E_\pi\left[\sum_{t=1}^{\tau_{max}} z_{i,t}\right] + E_\pi\left[\sum_{t=\tau_{max}+1}^{T}\sum_{k=1}^{z_{i,t}}\sqrt{\frac{1}{\sum_{s=1}^{t-\tau_{max}} z_{i,s}}}\right]$$

$$\le E_\pi\left[\sum_{t=1}^{\tau_{max}} z_{i,t}\right] + E_\pi\left[\sum_{t=\tau_{max}+1}^{T}\sum_{k=1}^{z_{i,t}}\sqrt{\frac{1}{\sum_{s=1}^{t-\tau_{max}} z_{i,s}}} - \sum_{t=\tau_{max}+1}^{T}\sum_{k=1}^{z_{i,t}}\sqrt{\frac{1}{\sum_{s=1}^{t-1} z_{i,s}+k}}\right.$$

$$\left. + \sum_{t=\tau_{max}+1}^{T}\sum_{k=1}^{z_{i,t}}\sqrt{\frac{1}{\sum_{s=1}^{t-1} z_{i,s}+k}}\right]$$

$$\le E_\pi\left[\sum_{t=1}^{\tau_{max}} z_{i,t}\right] + E_\pi\left[\sum_{t=\tau_{max}+1}^{T}\sum_{k=1}^{z_{i,t}}\sqrt{\frac{1}{\sum_{s=1}^{t-\tau_{max}} z_{i,s}}} - \sqrt{\frac{1}{\sum_{s=1}^{t-1} z_{i,s}+k}}\right]$$

$$+ E_\pi\left[\sum_{t=\tau_{max}+1}^{T}\sum_{k=1}^{z_{i,t}}\sqrt{\frac{1}{\sum_{s=1}^{t-1} z_{i,s}+k}}\right]$$

$$\le E_\pi\left[\sum_{t=1}^{\tau_{max}} z_{i,t}\right] + E_\pi\left[\sum_{t=\tau_{max}+1}^{T}\sum_{k=1}^{z_{i,t}}\frac{\sum_{s=t-\tau_{max}+1}^{t-1} z_{i,s}+k}{2(\sum_{s=1}^{t-\tau_{max}} z_{i,s})^{3/2}}\right]$$

$$+ E_\pi\left[\sum_{t=\tau_{max}+1}^{T}\sum_{k=1}^{z_{i,t}}\sqrt{\frac{1}{\sum_{s=1}^{t-1} z_{i,s}+k}}\right].$$

Since re-targeting duration is at most $D$, all users arriving before $t-D$ does not receive any further messages. It implies that $z_{i,t} \le D$. Thus,

$$E_\pi\left[\sum_{t=1}^{\tau_{max}} z_{i,t}\right] + E_\pi\left[\sum_{t=\tau_{max}+1}^{T}\sum_{k=1}^{z_{i,t}}\frac{\sum_{s=t-\tau_{max}+1}^{t-1} z_{i,s}+k}{2(\sum_{s=1}^{t-\tau_{max}} z_{i,s})^{3/2}}\right]$$

$$\le D\tau_{max} + C_6 D^2 \tau_{max} + D \le C_7 \tau_{max}.$$

We further have

$$E_\pi\left[\sum_{t=\tau_{max}+1}^{T}\sum_{k=1}^{z_{i,t}}\sqrt{\frac{1}{\sum_{s=1}^{t-1} z_{i,s}+k}}\right] \le C_8 E_\pi[\sqrt{T_i(T)}] \le C_8\sqrt{T},$$

where $T_i(T)$ denotes the number of message $i$ offerings before time $T$. Applying Lemma 3.1, we have

$$E_\pi\left[\sum_{t=1}^{T} 1(J_t^c)\right] \le \sum_{i \in X} E_\pi\left[\sum_{t=1}^{T} 1(B_{i,t}^c)\right] \le N \sum_{t=1}^{T} \frac{2}{t^4} \le C_9 N.$$

Combining all the results above, we have

$$\sum_{r=1}^{T} E_\pi[U(\mathbf{S}^*, \mathbf{v}, q(m^*))] - E_\pi[U(\mathbf{S}^r, \mathbf{v}, q(m_r))]$$

$$\leq \sum_{r=1}^{T} E_\pi[(U(\mathbf{S}^*, \mathbf{v}, q(m^*)) - U(\mathbf{S}^r, \mathbf{v}, q(m_r)))1(H_r)] + \sum_{r=1}^{T} E_\pi[1(H_r^c)]$$

$$\leq C(N + M^2)\sqrt{T \log T} + C'N\tau_{max}.$$

$\square$

# References

[1] Remco Van Der Hofstad. *Random graphs and complex networks*, volume 1. Cambridge university press, 2016.