# OpenReview forum: "Cascading Bandits: Optimizing Recommendation Frequency in Delayed Feedback Environments"
_NeurIPS.cc/2023/Conference — NeurIPS 2023 poster_

### Official Review · Reviewer_RjMs · 2023-07-05

**Soundness:** 3 good
**Presentation:** 3 good
**Contribution:** 3 good
**Rating:** 7
**Confidence:** 3

**Summary:**

This paper introduces several modifications to the cascading bandit setting to make it more suitable for settings where users can provide delayed feedback and abandon the system in response to receiving messages too frequently, such as online ad retargeting. It then provides an algorithm for solving the message queueing optimization problem given a fixed set of parameters describing the user/environment, then a UCB-style algorithm for learning the user/environment parameters while making decisions. It then provides regret bounds for its algorithm and provides experiments for the non-contextual and contextual setting.

**Strengths:**

Originality/Significance: The authors propose a modification to cascading bandits which allows for delayed feedback and user abandonment, which make it substantially more applicable to several problems such as re-targeting in online advertising.

Quality: The paper includes algorithms for the contextual and non-contextual cases, as well as a regret proof and experiments. Their proposed method is clearly supported by the evidence provided.

Clarity: Writing was generally clear.

**Weaknesses:**

1) The UCB heuristic seems to me like it may be mis-implemented in Algorithms 2 and 3, since it uses the UCB of q instead of the LCB. See question 1 for more details.
2) The experiments don't compare to a normal cascading bandits baseline.
3) If I understand correctly, each message can only be used once. While the setting is substantially more realistic than cascading bandits, this still seems like a restrictive assumption.

**Questions:**

1) Should Algorithms 2 and 3 call Algorithm 1 with q^LCB instead of q^UCB? The optimism in the face of uncertainty heuristic is based on always using upper bounds on how good something is, and it seems that the upper bound on $\gamma$ would be obtained using the lower bound on q instead of the upper bound. For example, in [1] the PrimalDualBwK Algorithm on page 23 uses the lower bound on resource consumption instead of the upper bound.
2) How important is the restriction to a fixed message frequency for a given message queue length? It seems that most things will email me more at first and then less later.
3) Is there also an assumption that the user delay is uncorrelated with the click probability? If so, it would be good to mention it since it seems plausible that users who ultimately click are likely to provide feedback sooner.

**Limitations:**

The limitation of potentially shifting user preferences was mentioned, though I would also re-iterate in the limitations section the uniform dissemination frequency mentioned earlier on in the paper.

This research seems to have similar societal impact to most other work on bandits applied to online advertising.

---

> ### Author Rebuttal · Authors · 2023-08-10
>
> We appreciate your commenting that our writing is generally good. Please find our point-by-point responses below.
>
> > Should Algorithms 2 and 3 call Algorithm 1 with $q^{LCB}$ instead of $q^{UCB}$? The optimism in the face of uncertainty heuristic is based on always using upper bounds on how good something is, and it seems that the upper bound on $\gamma$ would be obtained using the lower bound on q instead of the upper bound. For example, in [1] the PrimalDualBwK Algorithm on page 23 uses the lower bound on resource consumption instead of the upper bound.
>
> Thank you for this comment. Following your comment, we have added discussion to Section 3.1 of our paper to clarify the intuition of upper bound. Note that $q(m)$ is the probability of examining the next message, so if we optimistically estimate the sequence, we should use the upper bound of the resuming probability. Also, Lemma 3.2 has proved this property rigorously.
>
>
> > The experiments don't compare to a normal cascading bandits baseline.
>
> Thank you for this constructive comment. Following this comment, we have added experiments to compare our algorithm to two Thompson sampling-based algorithms: TS-Cascade for standard setting and LinTS-Cascade for personalized recommendation setting [2]. The two algorithms are proposed based on a normal cascading bandits model. We compare the performance of our proposed algorithms and baselines including Thompson sampling algorithms with the same setting described in our paper, and the results are shown in the two figures in the attached PDF file. In the experiments, our proposed UCB-like algorithms outperform the Thompson sampling-like algorithms, with the 57.32% reduction in regret by time T=30,000 for the non-contextual case in 100 independent runs and the 19.97% reduction in regret by time T=3,000 for the contextual case in 50 independent runs. Intuitively, our proposed algorithms are specialized to the modification setting in our paper, which is a generalization to the normal cascading bandits model. By contrast, the Thompson sampling-like algorithms are for the normal setting, and they don’t take the delayed feedback and user abandonment into consideration.
> Ref: [2] Thompson Sampling Algorithms for Cascading Bandits https://jmlr.csail.mit.edu/papers/volume22/20-447/20-447.pdf
>
> > If I understand correctly, each message can only be used once. While the setting is substantially more realistic than cascading bandits, this still seems like a restrictive assumption.
>
>  Thanks for your great comment. Our algorithm can also apply to the case where messages can be recommended repeatedly, with a modification of line 8 in Algorithm 2 or line 7 in Algorithm 3 that $O^r$ is not excluded from the candidate messages. Moreover, if we want to constrain the total number of repetitive displays of one message as $B$, we can create $B$ copies of that message in the candidate set.
>
> > How important is the restriction to a fixed message frequency for a given message queue length? It seems that most things will email me more at first and then less later.
>
> Thanks for your question. A fixed message frequency for a given message queue length is a simplified assumption in our model. The purpose is to capture the tradeoff between the total number of messages that can be disseminated and the probability of abandonment. If the frequency has changed during the recommendation process, it would be hard to evaluate the impact of frequency. Therefore, to ensure a straightforward and clear evaluation process, we fix the frequency for each user. We have added a short discussion in the revised paper.
>
> > Is there also an assumption that the user delay is uncorrelated with the click probability? If so, it would be good to mention it since it seems plausible that users who ultimately click are likely to provide feedback sooner.
>
>
> Thanks for your great comment. We do have such an assumption. We have added this assumption in the revised version of the paper.
>
> Thank you very much for your helpful review! We hope we have addressed your concerns satisfactorily.

---

> > ### Comment · Reviewer_RjMs · 2023-08-20
> >
> > Thanks for the thoughtful response! The additional experiments and answers to my questions have increased my confidence in the results, and I am revising my score upward.

---

### Official Review · Reviewer_MzDh · 2023-07-06

**Soundness:** 3 good
**Presentation:** 3 good
**Contribution:** 3 good
**Rating:** 4
**Confidence:** 3

**Summary:**

This work studied a generalization setting of the cascading bandit model which has been explored in existing literature. In detail, beyond the existing cascade model, it allows delayed feedback from the user, different number of pulled arms at different time steps, and takes the action frequency into consideration.
This work provides and analyzed an algorithm for the setting with delayed feedback and proposed another one for the setting with personalized recommendation.
Numerical experiments are presented to compare the performance of algorithms.



====================

I would like to keep the previous score after reading comments from other reviewers and the responses from authors.

**Strengths:**

1. This paper formulates a more realistic cascading bandit model. In detail, it generalized the existing model by taking delayed feedback, flexible length of recommendation list and action frequency into consideration.
3. The motivation to study this novel setting is well explained and convincing.
2. To help readers understand the formulation of the problem, Example 2.1 with Figure 1 provides a vivid instance.

**Weaknesses:**

1. The regret upper bound for Algorithm 2 is provided in Theorem 3.3, I think a detailed comparison to existing bounds should be provided here:
    1. I guess Algorithm 2 also works for the traditional cascading bandit setting, how will the regret bound be in the traditional setting? Does it match the state-of-the-art bounds?
    2. What is the corresponding lower bound in this novel setting? Are the upper/lower bounds tight?
2. What is the key challenge to derive a bound for Algorithm 3 in the setting with personalized recommendation?
3. In numerical experiments, Thompson sampling algorithms are not mentioned. I believe the authors should also compare Thompson sampling algorithms or explain the reason not to do so.
4. The literature review does not seem to be complete. I think the author(s) may consider to mention the following works on cascading bandits:
    1. Minimax Regret for Cascading Bandits https://proceedings.neurips.cc/paper_files/paper/2022/hash/bbb397739546de027bc81a2c2a8fb119-Abstract-Conference.html
    1. Thompson Sampling Algorithms for Cascading Bandits
https://jmlr.csail.mit.edu/papers/volume22/20-447/20-447.pdf
    1. Analysis of Thompson sampling for combinatorial multiarmed bandit with probabilistically triggered arms
https://proceedings.mlr.press/v89/huyuk19a.html

**Questions:**

Major concerns are listed in the *Weakness* section, and some possible typos are as below:
1. Line 26: 'multi-arm bandit' should be 'multi-armed bandit'
2. Line 81: 'our work extends the cascading bandits literature include' should be 'our work extends the cascading bandits literature by including'
3. Line 85: 'known a priori' should be 'known as a priori'

---

> ### Author Rebuttal · Authors · 2023-08-10
>
> We would like to sincerely thank you for your effort in reviewing our paper.
>
> > The regret upper bound for Algorithm 2 is provided in Theorem 3.3, I think a detailed comparison to existing bounds should be provided here:
> I guess Algorithm 2 also works for the traditional cascading bandit setting, how will the regret bound be in the traditional setting? Does it match the state-of-the-art bounds?
> What is the corresponding lower bound in this novel setting? Are the upper/lower bounds tight?
>
> Thanks for your question. Since the problem is built on a new model, typical ‘existing bounds’ do not exist. Following your suggestion, we compare our results with classical cascading bandits. Cascading bandit is a special case of our proposed model. [Tight Regret Bounds for Stochastic Combinatorial Semi-Bandits] has derived lower bound as $\Omega(\sqrt{NMT})$. Moreover, [Online Learning under Delayed Feedback] has shown the regret of $O(\sqrt{NTlog T}+N E[\tau])$. Our setup is more complicated since there are more unknown parameters that need to be learned (probability of abandonment), and the feedback is received in a delayed manner. The regret bound for our proposed algorithm is $O(\max(N,M^2)\sqrt{T}+N\tau)$. From the comparison, we can see that regret is optimal in $T$ as well as the delay duration. We have added this discussion in the revised paper.
>
> > What is the key challenge to derive a bound for Algorithm 3 in the setting with personalized recommendation?
>
> Thank you for this thought-inspiring question. Firstly, the analysis of contextual cascading bandits with delayed feedback shared similar challenges with non-contextual cascading bandits: 1) We allow dynamic content for active users who are remaining in the system, resulting in multiple interactions; 2) The feedback of individual messages may be delayed for a random amount of time and the frequency of the message dissemination across users could be different. Moreover, user feedback depends on both the choice of content and the dissemination frequency.
>
> Second, the contextual cascading bandits have additional challenges. In Algorithm 3, the parameters corresponding to clicking and leaving probability both need to be estimated, and their estimation error depends on several characteristics including $x$ and $w$. The parameters need to be estimated through supervised learning oracles for generalized linear function. In addition, because of the abandonment behavior, we can only observe partial feedback.
>
> > In numerical experiments, Thompson sampling algorithms are not mentioned. I believe the authors should also compare Thompson sampling algorithms or explain the reason not to do so.
>
> Thank you for this very constructive suggestion. Following your comment, we additionally implemented two Thompson sampling-based algorithms for the standard setting (TS-Cascade) and personalized recommendation setting (LinTS-Cascade) in the second paper you mentioned. We compare the performance of our proposed algorithms and benchmarks including Thompson sampling algorithms with the same setting described in our paper, and the results are shown in the two figures in the attached PDF file. In the non-contextual setting, we run each algorithm independently for 100 times, and the average regret of TS-Cascade at T=30,000 is 1289.01, with a 95% confidence interval of (1098.93, 1479.07). Thus, UCB outperforms TS-Cascade in our experiments. In the contextual setting, we run each algorithm independently for 50 times, and the average regret of LinTS-Cascade at T=3,000 is 367.26, with a 95% confidence interval of (331.74, 402.79), which is also worse than ContextualUCB.
> Meanwhile, although our results indicate that our two UCB-like algorithms outperform these benchmarks, we would like to clarify that the focus of this paper is to formulate a more realistic and generalized cascading bandit model and propose appropriate algorithms to solve it, as you mentioned in the strengths section.
>
> > The literature review does not seem to be complete. I think the author(s) may consider to mention the following works on cascading bandits:
>
> Thank you very much for pointing out these papers. We have added them in the section of literature review.
>
> > some possible typos are as below:
>    Line 26: 'multi-arm bandit' should be 'multi-armed bandit'
>    Line 81: 'our work extends the cascading bandits literature include' should be 'our work extends the cascading bandits literature by including'
>    Line 85: 'known a priori' should be 'known as a priori'
>
> Thanks for your comments. We have fixed these typos.
>
> Thank you very much for your helpful review! We hope we have addressed your concerns satisfactorily.

---

> > ### Comment · Reviewer_MzDh · 2023-08-20
> > **Thanks for your response**

---

> > > ### Author Response · Authors · 2023-08-21
> > >
> > > Thank you so much for all your valuable time reviewing our paper.
> > >
> > > Over the past two weeks we made the following extra revisions: 1) We have implemented two Thompson sampling-based algorithms for comparison. The result is shown in the attached PDF file, which indicates that our UCB-based algorithms can outperform the cascading bandits baselines. 2) We have completed literature review, including Thompson sampling algorithms and the latest upper and lower bounds of cascading bandits regret. 3) We have added further discussion of the regret of Algorithm 2, which shows that our algorithm can give a consistent result with existing algorithms in extreme cases, so we believe that our results are highly credible. 4) We have reorganized the assumptions of our model, which may help resolve misunderstandings of our setting.
> > >
> > > We sincerely hope that with these extra revisions, our paper could earn a higher score from you and be closer to acceptance. Thank you so much!

---

### Official Review · Reviewer_TbT8 · 2023-07-06

**Soundness:** 3 good
**Presentation:** 4 excellent
**Contribution:** 2 fair
**Rating:** 6
**Confidence:** 3

**Summary:**

This work introduces a new variant of cascading bandit that differs with the classical cascading bandit in the following ways: (1). the user might abandon the system with certain probability depending on the list length; (2). the list length is some variable to be optimized (even could be contextualized), instead of fixed in advance; and (3). the list could be adaptively updated as more user feedbacks coming in. This new variation is closer to real-world settings such as notification/personalized recommendation systems.

The paper proposes a dynamic programming approach for finding the optimal length and optimal list periodically and extend it to the contextualized setting where the relevance and abandon probability is a function of contextual features. Some simple synthetic experiments verify the effectiveness of the method compared with epsilon-greedy based approach.


**Strengths:**

- The setting this paper studies, extending from classical cascading bandit, is closer to real-world applications. The modeling/assumptions are realistic and not deviate that much more from the use-cases. I do like the new formulation.

- The paper provides theoretical understanding of the non-contextualized case, the dynamic programming approach is intuitive and interesting, the regret compared with the optimal list and list length is given. The paper further extends it to the non-contextualized case, allowing parametric form of the relevance and abandon probability. The method is solid and the analysis seems valid.

- The paper is very well-written. It motivates the problem in a crystal clear way and thoroughly listed the difference compared with the previous approach. I enjoy reading the paper.

**Weaknesses:**

- One thing i am concerned about is the computational feasibility of the method. It seems for each user, the exact DP is performed, which might be super expensive when the length list $m$ is large. Could the authors comment more on the computational aspect?

- For the experiments, the author mentioned that the features is based on the real-world dataset, but the model parameters are being simulated. I am wondering how the real-world features help here? It seems to me closer to a fully synthetic set-up. Could the authors comment more on this aspect?

- Following this, it seems the experiment is performed in a very small scale. In real-world scenarios, the number of available messages might be very large, and also the length of the list. Could the authors compare the proposed method in a larger scale experiments, preferably more real-world scenarios?

- It seems there are also RL based approach for solving the notification problem [1], have the authors thought of this aspect and it would be great to add literature review on this aspect and discuss the pros/cons, even empirical comparison?

Ref: [1]: https://arxiv.org/abs/2202.03867

**Questions:**

- Do we need to have a fixed list of messages? How the algorithms adapted to the new possible messages that arrive at different time-steps?

- One thing confused me is that whether the learned list length $m$ is updated in each DP? or it is only computed and fixed on the first round?

---

> ### Author Rebuttal · Authors · 2023-08-10
>
> Thank you very much for your great comments! Please see our point-by-point response below.
>
> >One thing I am concerned about is the computational feasibility of the method. It seems for each user, the exact DP is performed, which might be super expensive when the length list $m$ is large. Could the authors comment more on the computational aspect?
>
> Thank you for your comment. Instead of enumerating all possible lists, the computational complexity of our basic DP (Alg.1)  is polynomial in $m$. Consequently, Alg. 2 should also operate in polynomial time for each time step. Therefore, we believe that the computational feasibility is acceptable in practice. Even in scenarios where multiple users enter our system simultaneously, they will share the same optimal message sequence. Thus, this won't significantly increase the computational complexity when applied to a real recommender system. In addition, in numerical trials, we found the computational efficiency of our algorithm superior to many others which need to enumerate possible lists or combinations.
>
> > For the experiments, the author mentioned that the features are based on the real-world dataset, but the model parameters are being simulated. I am wondering how the real-world features help here? It seems to me closer to a fully synthetic set-up. Could the authors comment more on this aspect?
>
> Thanks for your question. We estimate parameters from the real-world dataset. To be more specific, in our experiment, we first estimate $q(1)$ from the difference between the proportion of single recommendations and clicking actions of the first message among all records in the AmEx User Click dataset. Then we take the form of $q(m)=\frac{1-\exp(-km)}{1+\exp(-km)}$. With respect to attraction probabilities, we find that real datasets often have a small v value (with a mean of 0.06 by K-S normality test, p = 0.033). Under this setup, our algorithm performs better than other benchmark algorithms.
>
> > Following this, it seems the experiment is performed in a very small scale. In real-world scenarios, the number of available messages might be very large, and also the length of the list. Could the authors compare the proposed method in a larger scale experiments, preferably more real-world scenarios?
>
> Thank you very much for this constructive comment. The real dataset has about 75 messages in total. Based on your comment, we have run the experiment of 1000 messages and found out that our UCB-like algorithm has the ability to reduce regret by 22.71%, compared to the best baseline EpsilonGreedy. Therefore, the scale of our experiment is no less than realistic cases from an order-of-magnitude point of view. As discussed in our response to Weakness 1, our proposed algorithm has the potential of performing well when dealing with large-scale data. The performance of the experiments with large-scale data demonstrates the practical value of our proposed algorithm.
>
> > It seems there are also RL based approach for solving the notification problem [1], have the authors thought of this aspect and it would be great to add literature review on this aspect and discuss the pros/cons, even empirical comparison?
>
> Thank you for pointing out the reference using the offline RL-based approach. We have added a brief discussion of RL related literature in the literature review. The problem that we studied can be potentially modeled as an RL problem, but the learning complexity would be much higher.
>
> > Do we need to have a fixed list of messages? How the algorithms adapted to the new possible messages that arrive at different time-steps?
>
> Thanks for your insightful question. It is not necessary to have a fixed list of messages. Because the attraction probabilities of messages are estimated at each time step, we can always recompute the optimal sequence length and recommend the message with the highest UCB among all the messages available, without losing effectiveness when new messages show up.
>
> > One thing confused me is whether the learned list length $m$  is updated in each DP? or it is only computed and fixed on the first round?
>
> Thanks to your comment, we added more description about the model setting to clarify that the list length $m$ is fixed for a single user and computed when a user enters the system, while $m$ is updated over time and might be different for users entering the system at different times. For each user, the learned list length is only computed and fixed during the first round. The algorithm can potentially be implemented in a way that the total number of messages sent to a single user is updated at each time step. However, if the frequency has changed during the recommendation process, it would be hard to evaluate the average frequency. Therefore, to ensure a straightforward and clear evaluation process, we fix the list length at the very beginning for each user.
>
> Thank you very much for your helpful review! We hope we have addressed your concerns satisfactorily.

---

> > ### Comment · Reviewer_TbT8 · 2023-08-15
> > **Thanks for the response!**
> >
> > Thanks for the authors' response, and I believe my initial score appropriately reflects the quality of this work.

---

### Official Review · Reviewer_GPGp · 2023-07-08

**Soundness:** 3 good
**Presentation:** 4 excellent
**Contribution:** 3 good
**Rating:** 6
**Confidence:** 4

**Summary:**

The paper considers a bandit setting where the agent selects a list of messages to be sent one at a time in a time window of length $D$. For each message, the user may be attracted (and sent a reward) or not (in which case she may leave the system with a probability that depends on the frequency of the messages).

The authors propose an algorithm to identify the optimal sequence of messages (including their size) if the parameters are known, and use it to derive two UCB-like algorithms dedicated to this setting. The first algorithm treats the messages and users as independent entities, while the second is dedicated to the contextual setting, where the attraction probabilities depend on known characteristics of the messages and the abandonment probability depends on known characteristics of the user.

The regret of the first algorithm is theoretically bounded, and some experiments compare both algorithms and epsilon-greedy.

**Strengths:**

The main strengths of the paper are the proposed setting and the derivation of the optimal solution for this setting when the parameters are known.
Regarding the setting, the notion of delayed feedback is not new. However, (i) the sending of messages one at a time, (ii) the dependence of the abandonment probability on the message frequency, and (iii) the distinction between attraction and reward, are interesting from an application point of view.

Two UCB-like algorithms are proposed, but they are simple.

Similarly, the experiences are interesting but remain of little impact since the setting is new and there is no baseline apart from $\epsilon$-greedy.

**Weaknesses:**

The theoretical analysis is limited to the worst-case regret and covers only the algorithm for independent arms.

Apart from this weakness,  I have a comment: The literature on click models includes a click model with probability of leaving the system: the *dependent click
model*. This model is used in the bandit setting in

> S. Katariya, B. Kveton, Cs. Szepesvári, and Z. Wen. DCM bandits: Learning to rank with multiple clicks. In Proceedings of the 33rd International Conference on Machine Learning, pages 1215–1224, 2016.

or by generic click-models algorithms such as TopRank (cited in the appendix) or UniRank

> Camille-Sovanneary GAUTHIER · Romaric Gaudel · Elisa Fromont. UniRank: Unimodal Bandit Algorithms for Online Ranking. ICML'22


**Questions:**

Do you know how to derive an instance-dependent proof  (and a proof of the corresponding lower-bound) ?

**Limitations:**

There is no section dedicated to the limits of the proposed approach.

---

> ### Author Rebuttal · Authors · 2023-08-09
>
> Thank you very much for commenting that our paper is technically solid. We answer your questions below:
>
> >Apart from this weakness, I have a comment: The literature on click models includes a click model with probability of leaving the system: the dependent click model.
>
> Thanks for your comments on the references. We have added the two papers that you mentioned in the section of literature review in the revised paper.
>
> >Do you know how to derive an instance-dependent proof (and a proof of the corresponding lower-bound) ?
>
> Regarding the instance-dependent upper bound and lower bound, we can derive the result similarly following Theorem 2 and Theorem 4 in the paper [Cascading bandits: Learning to rank in the cascade model], combined with the technique that we have used in the paper to deal with multiple interactions and the delayed feedback. We have added the comment to the paper, but due to the space limit, we mainly focus on the instance-independent regret.
>
> Cascading bandit is a special case of our proposed model. [Tight Regret Bounds for Stochastic Combinatorial Semi-Bandits] has derived lower bound as $\Omega(\sqrt{NMT})$. Moreover, [Online Learning under Delayed Feedback] has shown the regret of $O(\sqrt{NTlog T}+NE[\tau])$. Our setup is more complicated since there are more unknown parameters that need to be learned (probability of abandonment), and the feedback is received in a delayed manner. The regret bound for our proposed algorithm is $O(\max(N,M^2)\sqrt{T}+N\tau)$. From the comparison to two special cases, we can see that regret is optimal in $T$ as well as the delay duration. We have added this discussion in the revised paper.
>
> Thank you very much for your helpful review! We hope we have addressed your concerns satisfactorily.

---

> > ### Comment · Reviewer_GPGp · 2023-08-18
> > **Thanks for the rebuttal**
> >
> > Thank you for your rebuttal, which reinforces my opinion of this paper.

---

### Author Rebuttal · Authors · 2023-08-10

We thank all reviewers for their time and insightful comments to improve our paper. We appreciate the positive feedback. We provide clarifications below.

---

### Decision · Program_Chairs · 2023-09-21

**Decision:**

Accept (poster)

**Comment:**

This paper studies a generalization of cascading bandit. In particular, it considers a setting that allows delayed feedback and different numbers of pulled arms at different time steps, and takes the action frequency into consideration. This paper proposes and analyzes algorithms for the considered problem, and demonstrates some experiment results.

Most reviewers recommend to accept this paper. After reading the paper, the reviews, and the rebuttals, I think the strengths of this paper outweigh its weaknesses, and recommend accepting this paper as a poster.